METHODS

# Uncertainty-aware traction force microscopy

**Adithan Kandasamy**[1,2,3], **Yi-Ting Yeh**[1,2], **Ricardo Serrano**[3], **Mark Mercola**[3], **Juan C. del Alamo**[1,2,4]*

**1** Department of Mechanical Engineering, University of Washington, Seattle, Washington, United States of America, **2** Center for Cardiovascular Biology, University of Washington, Seattle, Washington, United States of America, **3** Cardiovascular Institute and Department of Medicine, Stanford University, Stanford, California, United States of America, **4** Division of Cardiology, University of Washington, Seattle, Washington, United States of America

* juancar@uw.edu

**Data availability statement:** All relevant data are within the paper, its supporting Information files, and online at https://github.com/adithank/TFM-UQ.

## Abstract

Traction Force Microscopy (TFM) is a versatile tool to quantify cell-exerted forces by imaging and tracking fiduciary markers embedded in elastic substrates. The computations involved in TFM are often ill-conditioned, and data smoothing or regularization is required to avoid overfitting the noise in the tracked displacements. Most TFM calculations depend critically on the heuristic selection of regularization (hyper-) parameters affecting the balance between overfitting and smoothing. However, TFM methods rarely estimate or account for measurement errors in substrate deformation to adjust the regularization level accordingly. Moreover, there is a lack of tools for uncertainty quantification (UQ) to understand how these errors propagate to the recovered traction stresses. These limitations make it difficult to interpret the TFM readouts and hinder comparing different experiments. This manuscript presents an uncertainty-aware TFM technique that estimates the variability in the magnitude and direction of the traction stress vector recovered at each point in space and time of each experiment. In this technique, a non-parametric bootstrap method perturbs the cross-correlation functional of Particle Image Velocimetry (PIV) to assess the uncertainty of the measured deformation. This information is passed on to a hierarchical Bayesian TFM framework with spatially adaptive regularization that propagates the uncertainty to the traction stress readouts (TFM-UQ). We evaluate TFM-UQ using synthetic datasets with prescribed image quality variations and demonstrate its application to experimental datasets. These studies show that TFM-UQ bypasses the need for subjective regularization parameter selection and locally adapts smoothing, outperforming traditional regularization methods. They also illustrate how uncertainty-aware TFM tools can be used to objectively choose key image analysis parameters like PIV window size. We anticipate that these tools will allow for decoupling biological heterogeneity from measurement variability and facilitate automating the analysis of large datasets by parameter-free, input data-based regularization.

**Funding:** AK, YTY and JCdA have been partially supported by the US National Institutes of Health under grant numbers (1R01HD109149-73701, 1R01AI167943-01A1, 1R01HL170607-01A). RS and MM have been partially supported by the US National Institutes of Health under grant numbers (5R33HL167258-02,5R01HL152055-03).The funders had no role in study design, data collection and analysis, decision to publish, or preparation of the manuscript.

**Competing interests:** The authors have declared that no competing interests exist.

## Author summary

The ability to measure the cell-exerted mechanical forces on the surrounding substrate has led to fundamental and translational advancements in cell biology. Traction Force Microscopy (TFM) is a semi-computational method that tracks substrate deformation using fluorescent markers and back calculates the forces that give rise to the imaged displacements. Regularization is required to prescribe a degree of smoothness in the recovered forces that avoids overfitting the noise in experimental data. However, there is a lack of tools to objectively select the level of regularization based on input data quality. To overcome these limitations, we present an uncertainty-aware traction force measurement method (TFM-UQ) that adapts the level of smoothing locally according to image and motion-based errors providing the variability of traction stress read-outs. TFM-UQ removes the need for explicit user-selected regularization parameter, provides information to distinguish biological heterogeneity from measurement variability and is attractive for automatic analysis, and quality control of large datasets for high-throughput experiments.

## 1. Introduction

Biomechanical force measurement tools in the range of pico and nano Newtons [1–4] have been implicated in fundamental cellular functions such as cell fate determination [5], cell positioning during embryonic development [6], individual and collective cell migration [7], and cardiac contractility [8]. Fluorescence microscopy-based methods such as traction force microscopy (TFM) [9–13], Förster resonance energy transfer (FRET) [14], and DNA-hairpin-based sensors [15,16] provide spatial maps of traction stress and force fields with cellular and sub-cellular spatial resolution, and high temporal resolution. Among these techniques, TFM has become a widely used tool to measure cell-exerted traction forces by tracking fiduciary markers (e.g., fluorescent beads) embedded in linearly elastic substrates of known mechanical properties.

Typical TFM experiments do not require highly specialized laboratory equipment, and their preparation is relatively simple. In particular, two-dimensional (in-plane) and three-dimensional (out-of-plane) TFM experiments on flat surfaces can be performed with wide-field fluorescence microscopes [17], and image analysis software is widely available and runs in near real-time with off-the-shelf computers [18–21]. There is a vast body of scientific literature using TFM in biomechanics and mechanobiology studies. In particular, TFM has been instrumental in investigating single cell locomotion in planar surfaces [9,22], focal adhesion dynamics in adherent cells [23,24], intercellular forces in collectively migrating epithelial monolayers [25–28], human cardiomyocyte contractility [29,30], platelet spreading [31,32], leukocyte transmigration and interstitial migration [33–36], and host-pathogen interaction [37].

TFM is a semi-computational method that solves the inverse elastostatic problem of finding the traction stress field which gives rise to the observed field of substrate displacements. This inverse problem is ill-posed because of the long-range decay of displacements, making the calculated traction stresses highly sensitive to experimental measurement noise of high spatial frequency [4,38,39]. The ill-posedness of TFM manifests itself in the numerical discretization of the elastostatic problem, leading to a linear system of equations with a high condition number [9,23]. Even when this numerical issue is avoided by sparsely block-diagonalizing the elastostatic system and inverting it exactly in Fourier space [10,11], the slow

decay of substrate deformation still causes noise amplification. Therefore, TFM algorithms have traditionally included regularization schemes in the form of implicit priors [9,21,38–40], explicit noise filters [23,28], etc. Previous works have been instrumental to understand the differences among TFM methods [21,23,41]. However, there is no unified framework to comprehensively study the sensitivity and variability of TFM methods to the choice of elastostatic solution, numerical discretization, and the form of regularization scheme.

Most regularization methods in TFM involve optimizing a composite norm weighing model accuracy (data fit) and complexity (e.g. smoothness). This optimization determines the degree of spatial smoothing applied in the inversion process, affecting the dynamical range and the effective resolution of the recovered traction stress maps [38,39,42]. Several approaches have been introduced to balance these effects, e.g., $\chi^2$-Criterion [9,39], Morozov discrepancy principle [39,43], the self-consistency principle [39], the L-curve criterion [23,39], etc. Recent efforts have adopted Bayesian frameworks [21,44,45] following the seminal work of [9], proposing statistical metrics such as generalized cross-validation [23,41,46] to tailor the regularization level to each experimental deformation map. However, the optimal value of the regularization parameter estimated by the different criteria for the same experimental data has not always been consistent, leaving room for ambiguity in choosing the level of regularization [39].

Optimization metrics for TFM regularization are usually calculated over an entire microscopy image or region of interest, assuming that the noise is spatially uniform (i.e., homoskedastic). Therefore, the ensuing regularization is sensitive to localized anomalies (e.g., large bead aggregates and empty areas without beads). This sensitivity generates the need to introduce quality control steps in TFM analysis pipelines, which are usually binary (i.e., accept vs. reject) and seldom reported, thus limiting the yield and reproducibility of TFM experiments. More generally, fluorescence microscopy images for TFM can contain spatial gradients in bead focus or brightness, leading to spatially dependent noise in the measured substrate deformations. However, regularization of the inverse elastostatic problem rarely accounts for these spatial gradients. A recent work reformulating the optical flow method using the elastostatic equation as an optimization constraint is an important step towards effectively linking spatially-dependent image noise and the recovered traction stresses [43].

The vast majority of regularization schemes in TFM do not provide *a posteriori*, point-by-point estimates of experimental uncertainty. A usual approach to make up for this lack is to perform repeat experiments and report the statistics (e.g., mean and standard deviation) of select readouts. However, this approach mixes the measurement uncertainty that is uniquely due to the experimental setup, imaging method, and mathematical model with the biological variability associated with the fact that two different cells do not exert the same traction stresses. The paucity of uncertainty quantification (UQ) tools in TFM is at odds with the considerable UQ efforts in particle image velocimetry (PIV) and other common methods to measure fluorescent bead displacements [47–51]. Consequently, the choice of crucial parameters affecting the resolution and noise in the measured displacements for TFM, e.g., the PIV interrogation window size, must be done without quantitative information about this choice's effect on the traction stresses. This factor adds to the ambiguity in regularization criteria, sensitivity to image artifacts, and impossibility to separate biological and experimental uncertainty, perpetuating the view that TFM is technically involved and challenging to interpret [42,52], hindering its more widespread adoption.

This manuscript presents a traction force microscopy method with uncertainty quantification (TFM-UQ). The new TFM-UQ method adaptively regularizes the traction stresses in different regions of a measurement image to be self-consistent with the input deformation noise. Furthermore, it propagates the input noise forward to provide spatial uncertainty maps

in the recovered traction stresses. The input noise in TFM-UQ is determined using a novel, non-parametric PIV uncertainty quantification (PIV-UQ) method suitable for fiduciary markers used in cell mechanics experiments. A computationally efficient PIV software with GPU acceleration is developed to make the PIV-UQ method tractable. To validate the TFM-UQ pipeline, we performed numerical TFM experiments that directly simulated the sources of image noise with synthetically generated fluorescent beads. In experimental microscopy images with different cell types (including images with visible imaging artifacts), the TFM-UQ method uncovered spatially heterogeneous PIV measurement noise, uncertainty associated with the choice of image processing procedure, and finally, provided confidence intervals to the traction stress output. Taken together, TFM-UQ improves the dynamic range of traction stresses and provides a tool to decouple biological heterogeneity from measurement (and procedural) variability. This framework is attractive for enabling reproducible TFM results and automating quality control for analysis of large datasets. TFM-UQ code (Matlab) is open-source and is available at https://github.com/adithank/TFM-UQ.

## 2. Methods

The methods section is organized as follows. First, we provide an overview of traditional traction force microscopy (TFM) and introduce the workflow used in TFM with uncertainty quantification (TFM-UQ, §2.1). Then, we describe a new PIV method to measure substrate deformation and estimate its uncertainty (§2.2), a crucial input to TFM-UQ. Next, we describe the elastostatic model used in this work (§2.3). Finally, we describe a hierarchical Bayesian framework (§2.4) to propagate the uncertainty from deformation to traction stress and locally adjust regularization to account for spatially varying uncertainty levels.

### 2.1. Overview of TFM and TFM-UQ

This section is intended to provide a high-level summary of TFM-UQ for the potential users of the technique, while the rest of the methods section offers the technical details of the implementation. Traction force Microscopy (TFM) requires a snapshot of fiduciary substrate deformation markers taken under stress, i.e., $I_t$, to be compared with a stress-free reference image, $I_{\text{ref}}$. The stress-free state is achieved by lysing the cells, waiting for cells to move out of the region of interest, or using prior knowledge about fiduciary marker positions. The set of images $I = \{I_{\text{ref}}, I_t\}$ form the experimental data for TFM analysis (Fig 1A). Traditional TFM is implemented in two independent steps (Fig 1B). The first step determines the point-wise most likely substrate deformation field, e.g., using Particle Image Velocimetry (PIV) ($u_{\text{PIV}}$), between $I_t$ and $I_{\text{ref}}$. The second step solves the inverse problem of recovering the traction stresses $t$.

Traditional TFM methods output a point estimate of the $u_{\text{PIV}}$ and $t$ fields as well as integrated quantities (Fig 1B). Each step of the pipeline contains analysis parameters that control the magnitude and effective resolution of the recovered stresses. These methods propagate noise from the images to the traction stresses, but the analysis pipeline at every step is not aware of this uncertainty or the effect of analysis parameters. Therefore, an estimate of the traction force error is unavailable in traditional TFM workflows.

Fig 1C summarizes the TFM-UQ analysis pipeline. In addition to measuring the deformation field, the deformation errors due to image quality are estimated with a bootstrap method. Since PIV is a common method to measure substrate deformation in TFM, PIV-UQ (see §2.2) is developed to perturb the input image pixels to find a model-free estimation of the empirical distribution of the deformation field conditional on the $I$ pair. Nevertheless, the TFM-UQ

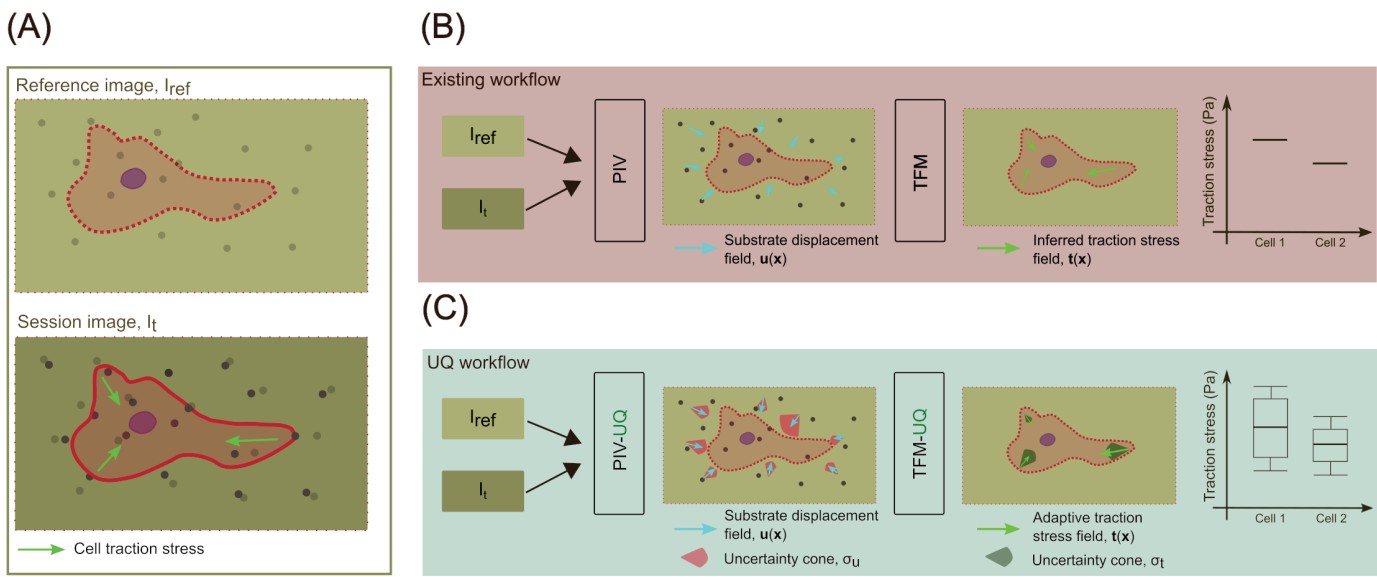

**Fig 1. TFM-UQ method improves traction stress inference and provides uncertainty bounds.** (**A**) Traction Force Microscopy experiment. Reference image ($I_{ref}$) of the substrate markers is obtained at a stress-free state. Session image(s) ($I_t$) of the substrate marker are obtained at the time point of interest. (**B**) Existing TFM workflow provides a point estimate of forces but does not consider the variability in the traction stress due to the microscopy images and TFM implementation. (**C**) The proposed TFM-UQ method considers the local image quality, ambiguity in regularization parameter selection, and the numerical implementation to provide locally adaptive smoothing and error bars.

pipeline is not critically dependent on the PIV method, and other registration techniques with UQ capabilities [51] could be adapted to work with TFM-UQ.

In the second step of the pipeline, the deformation vector ($\boldsymbol{u}_{\text{PIV}}$) and its covariance matrix quantifying uncertainty ($\boldsymbol{\Sigma}_{\text{PIV}}$) resulting from PIV-UQ are used as inputs to a Bayesian TFM inversion framework. We adopt a hierarchical Bayesian approach in which the model hyper-parameters, such as the level of regularization, are treated as random variables. This strategy allows us to regularize the inversion according to the noise in the deformation measurements introduced by the spatially varying features of each $\boldsymbol{I}$ pair. Finally, the hierarchical Bayesian approach provides the confidence interval for traction stress output (Fig 1C), quantifying how image quality and analysis parameters affect traction stress precision. This previously unavailable information is valuable for automated and objective quality control (e.g., useful when dealing with thousands of images), avoiding subjective selection of analysis parameters, or differentiating measurement variance from biological uncertainty. Lastly, analysis of traction stress and deformation variances from TFM-UQ allows users to interpret how the underlying algorithms balance the image data and required constraints (e.g., smoothing).

## 2.2. PIV uncertainty quantification (PIV-UQ) of substrate deformations

Particle image Velocimetry (PIV) is a commonly used method to measure substrate deformations from a pair of TFM images $\boldsymbol{I} = \{I_{\text{ref}}, I_t\}$. PIV identifies overlapping rectangular sub-domains $\mathbf{W}_{ij}$ (a.k.a. interrogation windows) in each member of $\boldsymbol{I}$, and registers them using cross-correlation as the similarity metric (Fig 2A). This operation is performed for each $\mathbf{W}_{ij}$ resulting in a discrete deformation field $\boldsymbol{u}_{\text{PIV}}$. Typically, PIV uses square interrogation windows of size $W_L$ (pixels) separated by a constant distance of $W_S$ pixels across the entire analyzed image. It is customary to set the window spacing as a fraction of window size, with

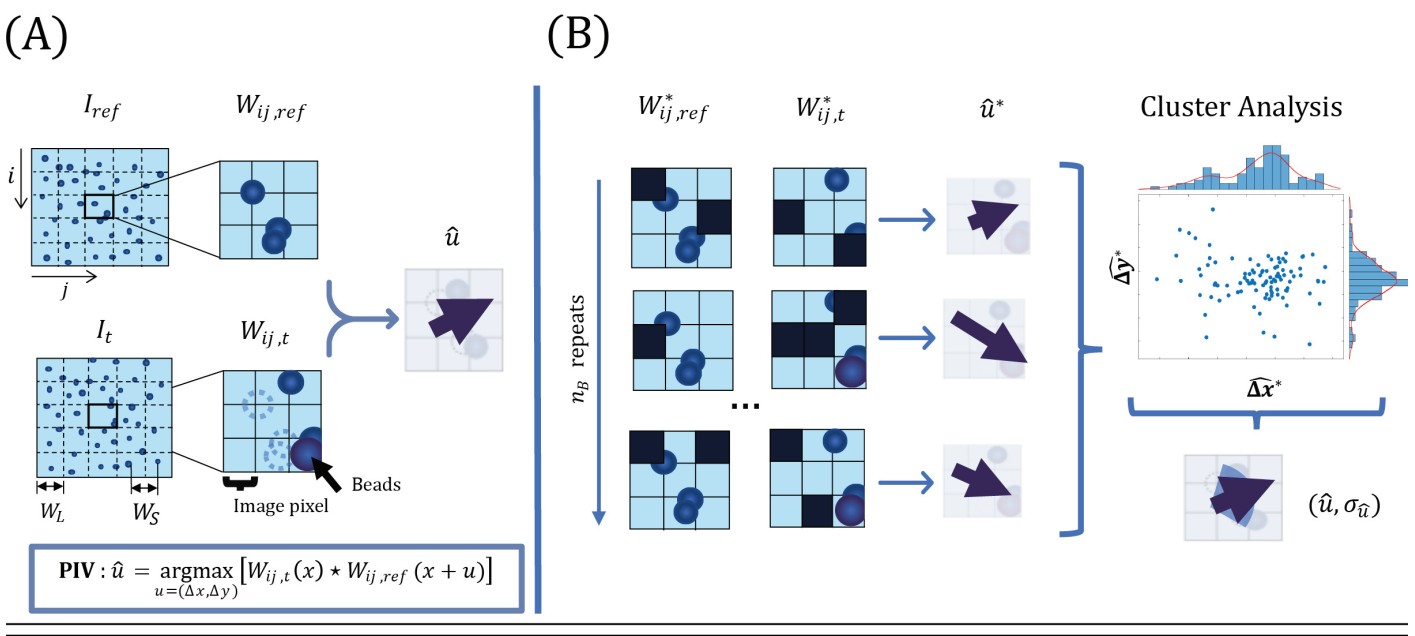

**Data:** Interrogation sub-windows, $\mathbf{W}_{ij} = \{\mathbf{W}_{ij,ref}, \mathbf{W}_{ij,t}\} \in \mathbb{R}^{n \times n}$
**Result:** $\hat{\mathbf{u}} \in \mathbb{R}^2, \sigma_{\hat{u}} \in \mathbb{R}^+$

$\hat{\mathbf{u}} \leftarrow \text{PIV}(\mathbf{W}_{ij,ref}, \mathbf{W}_{ij,t})$ ;
**for** $b = 1 : n_B$ **do**
 **for** $\mathbf{W}^*_{ij} = \{\mathbf{W}_{ij,ref}, \mathbf{W}_{ij,t}\}$ **do**
 Pixel indices, $i_p \leftarrow \text{U}(1, n^2, n^2)$;
 Deletion indices, $i_d \leftarrow [1 : n^2] \backslash i_p$;
 $\mathbf{W}^*_{ij}(i_d) \leftarrow 0$;
 **end**
 $\hat{\mathbf{u}}^*_b \leftarrow \text{PIV}(\mathbf{W}^*_{ij,ref}, \mathbf{W}^*_{ij,t})$ ;
**end**
$\sigma_{\hat{u}} \leftarrow \text{ClusterAnalysis}(\{\hat{\mathbf{u}}^*\})$ ;

**Fig 2. Particle Image Velocimetry with Uncertainty Quantification (PIV-UQ) method.** Top: Schematic of PIV and PIV-UQ. (**A**) The image data $I$ is sectioned into overlapping square sub-windows $\mathbf{W}_{ij}$ of length $W_L$ ($n^2$ pixels), their centers separated by $W_S$. PIV computes the most likely displacement for each sub-window $\mathbf{W}_{ij}$ that corresponds to the maximum of the cross-correlation metric. Each pixel of $\mathbf{W}_{ij}$ contributes to the correlation value, hence the maximization procedure. (**B**) PIV-UQ method bootstraps the contribution of individual pixels to the cross-correlation metric. Pixel indices are randomly sampled with replacement and the unsampled indices are set to 0 (Black pixels). Backslash denotes the set difference operation and $\text{U}(1, n^2, n^2)$ represents uniformly distributed $n^2$ samples in the interval $[1, n^2]$. The bootstrapped sample of size $n_B$ is analyzed for multiple possible clusters and outliers, subsequently providing a metric of variability for each sub-window (i.e., for each discrete vector of the deformation field). Bottom: PIV-UQ bootstrap algorithm.

$W_S = W_L/2$ providing the optimal trade-off between computational cost and the effective spatial resolution based on the Nyquist principle. Consequently, the error and resolution of PIV are usually dictated by one parameter, i.e., $W_L$. Since, for a given image pair, both the PIV resolution and error increase with $W_L$, this parameter must be chosen carefully, but current PIV methods used for TFM do not provide information about the uncertainty in $\boldsymbol{u}_{\text{PIV}}$. Below, we introduce a PIV-UQ method that estimates this measurement uncertainty. It can be used independently of TFM-UQ to inform the choice of parameters for PIV analysis of experimental images or in conjunction with TFM-UQ to propagate measurement uncertainty to the recovered traction stresses.

PIV-UQ estimates the most likely substrate displacement $\hat{\mathbf{u}}$ and its uncertainty, $\sigma_{\hat{u}}$ by perturbing the input images to obtain a non-parametric, model-free approximation of the empirical distribution of substrate displacements given the images $\boldsymbol{I}$. PIV-UQ applies the

bootstrap procedure outlined in Fig 2 (panel B and algorithm). Bootstrapping is a computational method based on random resampling of limited empirical data to construct measures of accuracy for sample estimates [53,54]. As described in the figure, the bootstrap method uses square interrogation windows. However, this method is generally applicable to any PIV implementation, including more advanced algorithms using iterative window refinement or window deformation, and the various sub-pixel interpolation methods for finding the correlation peak [55] by replacing the `PIV` function accordingly. A first nominal PIV run is carried out for a given pair of interrogation windows to obtain the estimator $\hat{\mathbf{u}}$ for the $\boldsymbol{u}_{\text{PIV}}$ field. Next, bootstrapped interrogation windows $\mathbf{W}_{ij}^*$ are built by randomly resampling pixels from $\mathbf{W}_{ij}$ with replacement, and unsampled pixels are left zero valued. The number of unsampled pixels is randomly drawn between 1 and the total number of pixels in the window using a uniform distribution. Then, PIV is performed to obtain the perturbed estimator $\hat{\mathbf{u}}^*$ from $\mathbf{W}_{ij}^*$. This process is repeated for $n_B$ trials so that the set of bootstrap estimators $\{\hat{\mathbf{u}}^*\}$ represent each interrogation window's empirical distribution $p(\hat{\mathbf{u}}|\mathbf{W}_{ij})$.

The bootstrapped distributions $p(\hat{\mathbf{u}}|\mathbf{W}_{ij})$ are further processed using density-based `clusterAnalysis` (DBSCAN algorithm [56] with parameters $\epsilon = W_S/5$ and `minPts` = $n_B/10$) to identify outliers from the bootstrap procedure and verify their uni-modality. The parameter `minPts` denotes the minimum number of points required to form a cluster (scaled with the number of bootstrap iterations $n_B$), and $\epsilon$ is the minimum distance between two points within a cluster (scaled with the spacing between PIV windows $W_S$). Because the PIV cross-correlation optimization looks for pure translations of $\mathbf{W}_{ij}$ in $\boldsymbol{I}$, motion patterns that significantly depart from translation (e.g., rotation, shear) can lead to multi-modal distributions where more than one value of $\hat{\mathbf{u}}$ is plausible [55]. In `clusterAnalysis`, $\epsilon$ is important to identify multi-modality and, hence, PIV invalidation when the minimum distance from edges of clusters are separated by at least 20% of the PIV window spacing. In addition, image noise, lack of image features, and brightness gradients can bias $p(\hat{\mathbf{u}}|\mathbf{W}_{ij})$ towards a uniform or skewed distribution, producing erroneous estimates of $\hat{\mathbf{u}}$. PIV-UQ utilizes $p(\hat{\mathbf{u}}|\mathbf{W}_{ij})$ to validate $\hat{\mathbf{u}}$ instead of purely heuristic-based metrics such as ratio of correlation peaks [55,57]. The "bad" windows that fail the validation are deleted, and the standard deviation for each of the remaining sub-windows ($\sigma_{\hat{u}}$) is reported . For the purposes of TFM-UQ, the standard deviation of these windows are set to a high number (e.g., a factor of 5 to the maximum $\sigma_{\hat{u}}$). The number of "bad" windows also serves as a diagnostic tool in selecting the parameter $W_L$ and validating the PIV results for the image $\boldsymbol{I}$. For the rest of the manuscript, $\boldsymbol{u}_{\text{PIV}}(\mathbf{x}) = \hat{\mathbf{u}}(\mathbf{x})$ and $\sigma_{u,\text{PIV}}(\mathbf{x}) = \sigma_{\hat{u}}(\mathbf{x})$ for each interrogation window centered at $\mathbf{x} = (x, y)$. Coefficient of variation (CoV) = $\text{Std}[\sigma_{\hat{u}}]/\text{Mean}[\sigma_{\hat{u}}]$ is used as a metric to assess convergence of $\sigma_{\hat{u}}$ with number of bootstrap iterations $n_B$ (see §3.3 and Fig D in S1 Text).

## 2.3. Elastostatic model

We consider a flat inertia-less, linearly elastic substrate of Young's modulus $E$ and Poisson's ratio $\nu$, deformed under the action of traction stresses exerted at its free surface $z = h$. The substrate deformation field ($\boldsymbol{u}$) satisfies the elastostatic partial differential equation (PDE)

$$\nabla^2 \boldsymbol{u} + \frac{\nabla(\nabla \cdot \boldsymbol{u})}{1 - 2\nu} = 0, \tag{1}$$

with boundary conditions $\boldsymbol{u}(x, y, z = 0) = \boldsymbol{0}$ and $\boldsymbol{u}(x, y, z = h) = \boldsymbol{u}_{\text{h}} \approx \boldsymbol{u}_{\text{PIV}}$. Once this boundary value problem is solved, the constitutive relation for the stress tensor $\mathbf{T} = F(x, y, z; \boldsymbol{u}, E, \nu)$ allows for computing the traction stress vector at $z = h$ as $\boldsymbol{t}(x, y) = \mathbf{T}(x, y, z = h) \cdot \boldsymbol{n}$, where $\boldsymbol{n} \approx (0, 0, 1)$ is the vector normal to the substrate surface. Because the constitutive equation

(i.e., $F$) and the elastostatic PDE are linear, we can denote the relationship between $t$ and $u_h$ concisely using the linear transformation $\mathbf{M} : t \rightarrow u_h$, where $\mathbf{M}$ is the elastic response operator. $u_h$ and $t$ can be two- or three-component vector fields, and $h$ can be finite or infinite.

Using this formulation, the forward problem of finding $u_h$ given a traction stress field $t$ can be written as

$$u_h = \mathbf{M}t + \varepsilon_\beta + \varepsilon_m, \tag{2}$$

$$\varepsilon_\beta \sim N(0, \beta^{-1}\mathbf{U}), \tag{3}$$

$$\varepsilon_m \sim N(0, \boldsymbol{\Sigma}_{\text{PIV}}). \tag{4}$$

where $\varepsilon_m$ is the measurement error arising from PIV and $\varepsilon_\beta$ represents the other error sources from the elastostatic model, numerical discretization, etc. Both error terms are modeled as normally distributed, but $\varepsilon_\beta$ is assumed to be global with precision parameter $\beta$, whereas $\varepsilon_m$ is characterized by the spatially dependent variance $\boldsymbol{\Sigma}_{\text{PIV}}$. Therefore, $\mathbf{U}$ is the identity matrix while $\boldsymbol{\Sigma}_{\text{PIV}}$ is a diagonal matrix. The precision parameter $\beta$ is unknown and is modeled as a hyperparameter in our hierarchical Bayesian approach (§2.4). On the other hand, $\boldsymbol{\Sigma}_{\text{PIV}}$ is measured directly using the PIV-UQ method described in the previous section (§2.2), and therefore, it will be treated explicitly.

The form of $\mathbf{M}$ depends on each model's geometry (e.g., finite vs. infinite $h$, 2D vs. 3D deformation), solution method (Fourier transform, finite elements, etc), and substrate material properties. The particular form of $\mathbf{M}$ is inconsequential for the TFM-UQ algorithm, as seen below. For simplicity, this report considers the Fourier Transform Traction Cytometry (FTTC) response function, $\widetilde{\mathbf{M}}$, to build $\mathbf{M}$. In the Fourier domain, $\widetilde{\mathbf{M}}$ adopts a straightforward block-diagonal form that can be derived analytically for finite and infinite $h$ for various 2D and 3D boundary conditions at the substrate's surface [10,11,58,59]. Without loss of generality, we used Lin *et al.*'s [58] concise derivation for the finite-$h$ case with 2D boundary conditions:

$$\widetilde{M}(\mathbf{k}) = g_1(k)\begin{bmatrix} 1 & 0 \\ 0 & 1 \end{bmatrix} + g_2(k)\begin{bmatrix} k_x^2 & k_xk_y \\ k_xk_y & k_y^2 \end{bmatrix}, \tag{5}$$

where, $g_1 = 1/f_1$, $g_2 = -f_2/f_1(f_1 + k^2f_2)$ and $k = |\mathbf{k}|$ is the absolute value of the wavenumber vector $\mathbf{k} = (k_x, k_y)$. The scalar functions are defined as

$$f_1 = E\frac{ck}{2(1+\nu)s}, \quad f_2 = E\frac{(3-4\nu)\nu sc^2 - (1-\nu)ckh + (1-2\nu)^2 s + 2(kh)^2}{(2(1-\nu^2)ks)((3-4\nu)sc + kh)},$$

with $c = \cosh kh$ and $s = \sinh kh$. Inverse Fourier transforming $\widetilde{\mathbf{M}}$, we obtain $\mathbf{M} = \mathbf{F}^H\widetilde{\mathbf{M}}\mathbf{F}$, where $\mathbf{F}$ is the discretized Fourier Transform (DFT) matrix and $\mathbf{F}^H$ its Hermitian transpose.

The finite-dimensional matrix $\mathbf{M}$ has a large condition number because the eigenvalues of $\widetilde{\mathbf{M}} \sim h$ for $kh \rightarrow 0$ (pure shear limit) while they vary as $k^{-1}$ for $kh \gg 1$. Using the Nyquist criterion to determine the maximum wavenumber, $k_{max} = 2\pi/W_L$, we obtain $\kappa(\mathbf{M}) \sim h/W_L$, which worsens as the substrate becomes thicker and the spatial resolution becomes finer. Depending on the elastostatic model's geometry, the condition number of $\mathbf{M}$ can become even higher. For instance, under a more realistic assumption that the TFM fluorescent beads are located slightly under the substrate's surface, i.e., at $z = h - \Delta h$, the measured displacements are amplified by an additional factor $\exp(k\Delta h)$ [11] and, consequently,

$\kappa(\mathbf{M}) \sim h/W_L \exp(2\pi\Delta h/W_L)$. Therefore, the noise terms $\boldsymbol{\varepsilon}_\beta$ and $\boldsymbol{\varepsilon}_m$ are susceptible to amplification when equation 2 is inverted, and the level of amplification may vary significantly among TFM methods.

## 2.4. Hierarchical Bayesian TFM formulation to propagate PIV uncertainty

This section presents a Bayesian framework to invert the elastostatic problem in the presence of local measurement noise and global model noise. A three-level hierarchical framework, summarized in Fig 3 is formulated. Consistent with the elastostatic model presented in equations (2)–(4), the likelihood of the observed displacement given a traction stress field, $p(\boldsymbol{u}_h|\boldsymbol{t})$, is governed by the elastostatic matrix $\mathbf{M}$. The error due to PIV measurement of $\boldsymbol{u}_h$ is modeled explicitly based on the variance $\boldsymbol{\Sigma}_{\text{PIV}}$ maps obtained from bootstrap PIV-UQ, and the random global error is modeled as additive noise of variance $\beta^{-1}$, where $\beta$ is a hyperparameter. The likelihood of measured displacements, $p(\boldsymbol{u}_h|\boldsymbol{t},\beta,\boldsymbol{\Sigma}_{\text{PIV}})$, can be expressed as a multivariate Gaussian distribution for the model residual,

$$p = (2\pi)^{-N_w}|\Lambda|^{-1/2}\exp\left[-\frac{1}{2}(\boldsymbol{u}_h - \mathbf{M}\boldsymbol{t})^T\Lambda^{-1}(\boldsymbol{u}_h - \mathbf{M}\boldsymbol{t})\right], \tag{6}$$

Here, $\Lambda = \beta^{-1}\mathbf{U} + \boldsymbol{\Sigma}_{\text{PIV}}$, and this matrix's determinant is expanded as the product of its diagonal elements, $|\Lambda| = \prod_{i=1}^{2N_w}\left[\beta^{-1} + \Sigma_{ii}^{\text{PIV}}\right]$, where $\Sigma_{ii}^{\text{PIV}}$ occupies the $i$-th position in the diagonal of $\boldsymbol{\Sigma}_{\text{PIV}}$. The dimensions of the multivariate Gaussian, $2N_w$, is given by the total number of displacement measurements (i.e., the number of interrogation windows $N_w$ multiplied by the number of displacement components).

The prior distribution of the traction stress, $p(\boldsymbol{t}|\alpha,\mathbf{L})$, is based on a linear quadratic functional $\boldsymbol{t}^T\mathbf{L}\boldsymbol{t}$, leading to the zero-mean multivariate Gaussian distribution, *i.e.*,

$$p \propto \alpha^{N_w}\exp\left[-\frac{\alpha}{2}\boldsymbol{t}^T\mathbf{L}\boldsymbol{t}\right], \tag{7}$$

where $\alpha$ is a hyper-parameter. The dimensions of $\boldsymbol{u}_h$ and $\boldsymbol{t}$ need not be the same depending on the form of the solution matrix $\mathbf{M}$ (e.g. Boussinesq solution [39]). We used the discretized Laplacian operator for $\mathbf{L}$, establishing a Tikhonov low-pass filter of the traction stress field.

The hyperparameters $\alpha$ and $\beta$ control the influence of the prior constraint and model misfit on the result respectively during the inversion process. The case of $\boldsymbol{\Sigma}_{\text{PIV}} = 0$ and fixed hyperparameter values corresponds to the traditional (frequentist or Bayesian) approach, which is governed by the regularization parameter $\lambda = \alpha/\beta$ [21,23,39,60]. In our formulation, setting $\boldsymbol{\Sigma}_{\text{PIV}} = 0$ would be equivalent to neglecting the PIV measurement error or to assuming it is spatially uniform so it can be lumped into the model error $\beta$.

In the hierarchical framework, the hyperparameters $\alpha$ and $\beta$ are modeled as random variables, using hyperpriors derived from the Gamma distribution, $p(x|\theta,\phi) = \phi^\theta x^{\theta-1}\exp\{-\phi x\}/\Gamma(\theta)$, where the term in the denominator is the Gamma function. To keep the hyperpriors non-informative, the shape and rate parameters were fixed to constant values ($\theta = 1, \phi = 10^{-5}$). This hierarchical approach (Fig 3) allows for including an acceptable level of variability of the hyperparameters given the observed $\boldsymbol{u}_h$ distribution. Note that the marginal prior distribution $p(\boldsymbol{t}|\mathbf{L},\phi,\theta) = \int_\alpha p(\boldsymbol{t}|\alpha,\mathbf{L})p(\alpha|\phi,\theta)d\alpha$ is a Student-t distribution [61] due to the hierarchical treatment of hyperparameter uncertainty.

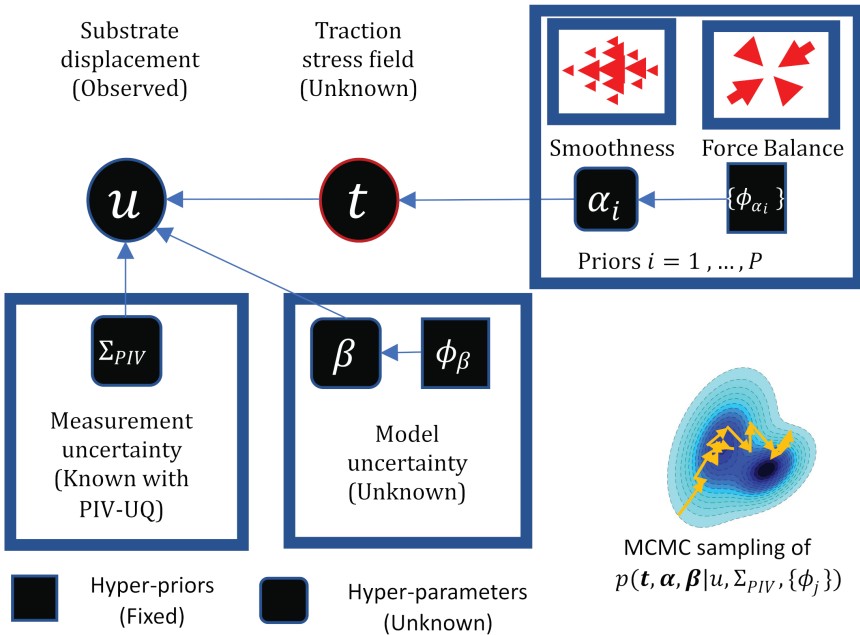

**Fig 3. Hierarchical Bayesian TFM formulation for adaptive and self-consistent regularization.** The problem of choosing a regularization parameter is treated in a Bayesian formulation. Substrate deformation ($u_h$) is the observed quantity, and it is modeled as the additive contribution of traction stresses ($t$), locally resolved PIV measurement uncertainty $\Sigma_{PIV}$ and a global model error ($\beta$) that is unknown. The hierarchical formulation allows to express unknown hyper-parameters ($\alpha, \beta$) for the prior on traction stress field and the model error term as random variables to be inferred. Therefore, a non-informative hyper-prior ($\phi_s$) is specified to the hyper-parameters. The priors can encode reasonable assumptions such as smoothness and global force balance. Markov Chain Monte Carlo (MCMC), specifically an hybrid Gibbs sampling, is used for inference from the marginal posterior distribution, $p(t|u_h)$.

The conditional posterior distributions $p(t|u_h, \alpha, \beta)$, $p(\alpha|t, u_h)$, and $p(\beta|t, u_h)$ were obtained from the joint distribution of the traction stress, displacement, and hyperparameters,

$$p(u_h, t, \alpha, \beta | \Sigma_{PIV}, \mathbf{L}) = p(u_h | t, \beta, \Sigma_{PIV}) p(t|\alpha, \mathbf{L}) p(\alpha|\phi_\alpha, \theta_\alpha) p(\beta|\phi_\beta, \theta_\beta) \propto$$

$$\propto |\Lambda|^{-1/2} \alpha^{N_w + \theta_\alpha - 1} \beta^{\theta_\beta - 1}$$

$$\exp\left\{ -\frac{1}{2} \left[ (u_h - \mathbf{M}t)^T \Lambda^{-1} (u_h - \mathbf{M}t) \right] - \frac{\alpha}{2} t^T \mathbf{L} t - \phi_\alpha \alpha - \phi_\beta \beta \right\}.$$

The posterior conditional distribution of traction stress is given by,

$$p(t|u_h, \alpha, \beta, \Sigma_{PIV}, \mathbf{L}) \propto \exp\left[ -\frac{1}{2}(u_h - \mathbf{M}t)^T \Lambda^{-1}(u_h - \mathbf{M}t) - \frac{\alpha}{2}t^T \mathbf{L}t \right]$$

$$\propto \exp\left[ -\frac{1}{2} \left( t - S_{post}\mathbf{M}^T \Lambda^{-1} u_h^T \right] S_{post}^{-1} \left[ t - S_{post}\mathbf{M}^T \Lambda^{-1} u_h^T \right) \right], \tag{8}$$

with the covariance matrix $\mathbf{S}_{post} = \left[ \mathbf{M}^T \Lambda^{-1} \mathbf{M} + \alpha \mathbf{L} \right]^{-1}$.

In turn, the conditional posterior distribution for the hyperparameter $\alpha$ is

$$p(\alpha|t, u_h, \mathbf{L}) \propto \alpha^{N_w + \theta_\alpha - 1} \exp\left[ -\frac{\alpha}{2} \left( 2\phi_\alpha + t^T \mathbf{L}t \right) \right]. \tag{9}$$

These two probability density functions correspond to a Gaussian and a Gamma distribution, i.e.,

$$t|u_{\mathrm{h}}, \alpha, \beta, \Sigma_{\mathrm{PIV}}, \mathbf{L} \sim N(\mathbf{S}_{\mathrm{post}}\mathbf{M}^T\Lambda^{-1}u_{\mathrm{h}}^T, \mathbf{S}_{\mathrm{post}}), \tag{10}$$

$$\alpha|t, \mathbf{L}, \theta_\alpha \sim \Gamma\left(N_w + \theta_\alpha, \frac{1}{2}|\mathbf{L}t|^2 + \phi_\alpha\right), \tag{11}$$

and can be found in closed form. However, a similar derivation for the conditional posterior distribution for $\beta$,

$$p(\beta|t, u_{\mathrm{h}}) \propto |\Lambda|^{-1/2}\beta^{\theta_\beta - 1}\exp\left[-\frac{1}{2}(u_{\mathrm{h}} - \mathbf{M}t)^T\Lambda^{-1}(u_{\mathrm{h}} - \mathbf{M}t) - \phi_\beta\beta\right], \tag{12}$$

did not produce an expression in closed form, since this expression also depends on $t$. Consequently, we used an iterative Markov Chain Monte Carlo (MCMC) method [62,63] to infer $p(t|u_{\mathrm{h}}, \alpha, \beta)$, $p(\alpha|t, u_{\mathrm{h}})$ and $p(\beta|t, u_{\mathrm{h}})$. The MCMC approach also provides a straightforward way to calculate the quantity of interest, i.e., the marginal posterior distribution of the traction stress field, $p(t|u_{\mathrm{h}}) = \int p(t|u_{\mathrm{h}}, \alpha, \beta)\mathrm{d}\alpha\mathrm{d}\beta$ by summing over MCMC draws. The statistics of interest are approximated from the MCMC draws after a burn-in period ($\mathrm{it_b}$). Specifically, the expected (mean) marginal posterior traction stress field is given by $\hat{t}(\mathbf{x}) = \mathbb{E}[t(\mathbf{x})|u_{\mathrm{h}}] \approx \frac{1}{N_d}\sum_i t(\mathbf{x})_i$ and the marginal posterior standard deviation of the traction stress is $\mathrm{Std}[t(\mathbf{x})|u_{\mathrm{h}}] \approx \{\frac{1}{N_d}\sum_i (t(\mathbf{x})_i - \mathbb{E}[t(\mathbf{x})|u_{\mathrm{h}}])^2\}^{1/2}$, where $t_i$ represents the $i^{th}$ draw of the vector of traction stress (out of $N_d$ draws) from the Hybrid-Gibbs sampler. $\hat{t}$ is the traction

---

**Data:** Displacement field, $u_{\mathrm{h}} \in \mathbb{R}^{2N_w}$; Displacement uncertainty, $\Sigma_{\mathrm{PIV}}$; $\alpha_0, \beta_0 \in \mathbb{R}_{>0}$; Burn-in iterations, $\mathrm{it_b}$, max iterations, $\mathrm{it_m}$

**Result:** Posterior samples : $\{t\} \in \mathbb{R}^{2N_w}$ ; $\{\alpha\}, \{\beta\} \in \mathbb{R}_{>0}$

Initialization;
$\theta_\alpha \leftarrow 1; \theta_\beta \leftarrow 1;$
$\phi_\alpha \leftarrow 10^{-5}; \phi_\beta \leftarrow 10^{-5};$
$\mathrm{it} \leftarrow 1;$
$\alpha \leftarrow \alpha_0; \beta \leftarrow \beta_0$
**while** $it < (it_b + it_m)$ **do**
 $\Lambda^{-1} \leftarrow [1/\beta\mathbf{U} + \Sigma_{\mathrm{PIV}}]^{-1};$
 $\mathbf{S}_{\mathrm{post}} \leftarrow [\mathbf{M}^T\Lambda^{-1}\mathbf{M} + \alpha\mathbf{L}]^{-1};$
 $\mathbf{S}_{\mathrm{post}} \leftarrow \frac{1}{2}[\mathbf{S}_{\mathrm{post}} + \mathbf{S}_{\mathrm{post}}^T];$
 $t \sim N(\mathbf{S}_{\mathrm{post}}\mathbf{M}^T\Lambda^{-1}u_{\mathrm{h}}, \mathbf{S}_{\mathrm{post}});$
 $\alpha \sim \Gamma\left(\frac{n}{2} + \theta_\alpha, \frac{1}{2}(t^T\mathbf{L}t) + \phi_\alpha\right);$
 /* Simulate $\beta$ by inverse transform sampling of discrete p.d.f */
 $\hat{\beta} \leftarrow \arg\min\log(p(\beta|u_{\mathrm{h}}, t, \theta_\beta, \phi_\beta));$
 Discrete $\beta$ posterior domain, $\Omega_\beta \in [\hat{\beta} - 3\sigma_{\hat{\beta}}, \hat{\beta} + 3\sigma];$
 c.d.f, $F(x) = \int_{\Omega_{\hat{\beta}}} p(\beta|u, t, \theta_\beta, \phi_\beta);$
 $\beta \sim F_x^{-1}(\mathtt{U(0,1)});$
 **if** $it > it_b$ **then**
 | Write $\{t, \alpha, \beta\}$.
 **end**
**end**

**Fig 4. Hybrid Gibbs sampler for TFM UQ inference.**

stress field output and the total, pointwise traction stress uncertainty $\sigma_t = \sqrt{\sigma_{tx}^2 + \sigma_{ty}^2}$, where $\sigma_x$ is $\text{Std}[t_x|\boldsymbol{u}_h]$ and so on.

A hybrid-Gibbs sampler was adopted to sample these distributions using equations (10–12), as described in Fig 4. The hyperparameters were initialized to $\alpha_0$ and $\beta_0$, which can be selected from reasonable maximum likelihood estimate values. In each sampling iteration, Gibbs sampling was performed by sequentially sampling the conditional posteriors, $p(\boldsymbol{t}|\boldsymbol{u}_h, \alpha, \beta)$ and $p(\alpha|\boldsymbol{t}, \boldsymbol{u}_h)$. The posterior $p(\beta|\boldsymbol{u}_h, \boldsymbol{t})$ was constructed numerically around the maximum *a posteriori* probability of $\beta|\boldsymbol{u}_h, \boldsymbol{t}$ and an inverse transform sampling using the numerical cumulative distribution function was performed for $\beta|\boldsymbol{u}_h, \boldsymbol{t}$. The burn-in was typically set to 20-50 iterations, and sampling was performed until convergence as assessed by trace plots and autocorrelation of the sampling chain. The convergence and, therefore, the computation time of Gibbs sampler depends on the size of the number of parameters ($N_w$), the strength of spatial correlation in prior (i.e. $\mathbf{L}$) and $\boldsymbol{\varepsilon}_m$.

Using a laptop computer with Intel Core i7-10875H CPU and < 10 GB RAM usage, PIV-UQ for a $2048 \times 2048$ px image analyzed with $W_L = 128 (N_w \approx 30)$ and $n_B = 50$ took about 19 seconds ($\approx 0.4$ seconds per PIV run). The PIV-UQ compute time depends on the size of the image and $W_L$ (e.g. with $W_L = 64$, 0.44 seconds per PIV run). For the typical resolution of the PIV field in TFM experiments, ($N_w \approx 30$ and $\mathtt{t}_m = 100$ iterations), it takes less than 3 minutes. For a higher resolution with $N_w \approx 60$, it takes approximately 1.5 hours. Since the matrix inversion is the computational bottleneck, workstations with more CPU cores can be used to speed up the computation.

## 2.5. Cell culture and imaging

Human vascular umbilical endothelial vein cells (HUVECs) (Cell Applications) were cultured in M199 (Gibco) supplemented with 10% (v/v) endothelial growth medium (Cell Applications), 10% (v/v) fetal bovine serum (Gibco), and 1% penicillin-streptomycin (Gibco). Mouse mesenchymal C3H/10T1/2 (ATCC) were cultured in low glucose Dulbecco's Modified Eagle's Medium (DMEM) supplemented with 10% v/v fetal bovine serum (Gibco) and 1% penicillin-streptomycin (Gibco). The cell plasma membranes were stained with CellMask (5 µg/mL, Thermo Fisher Scientific) to visualize the HUVEC cells. Imaging is acquired with a live-cell, wide-field microscope (Leica DMI8 S inverted), 40 X NA 1.3 oil immersion objective and a sCMOS camera (Leica DFC9000) using LAS X software (Leica). Background subtraction (as indicated) is performed with Leica Thunder software (Instantaneous computational clearing).

## 2.6. Micropatterned polyacrylamide gel preparation

3.5 mm glass bottom dishes were pretreated in an ultraviolet ozone (UVO) box for 5 min, and the glass surface was then activated by using 2 M NaOH (Sigma) for 5 min. The whole dishes were washed with distilled water, dried, and treated with 3-aminopropyl-trimethoxysilane (Sigma) for 20 min. After removing silane, glass surfaces were rinsed with 100% EtOH (Decon), dried, and treated with 0.5% glutaraldehyde (Sigma) for 30 min. The activated surfaces were rinsed with distilled water and kept at room temperature for use within the same day.

Round 12 mm glass coverslips were pretreated in a UVO box for 5 min. The coverslips were then incubated with a 110 µL drop of 0.2 mg/mL PLL-PEG (poly[L-lysine] grafted with poly[ethylene glycol]) (Nanosoftpolymers) for 30 min at room temperature. After 30 minutes, PLL-PEG was removed from the coverslip surface by aspiration. The chrome face of the photomask (Advance Reproductions) was activated by UVO for 3 minutes. The PLL-PEG-coated coverslips were attached to the chrome side of the photomask by sandwiching a 2 µL drop of

distilled water between both surfaces. The photomask and coverslips were then exposed to UVO light for 5 minutes. The photomask was detached from the coverslips by adding distilled water and then dried by aspirating excess water. A 110 µL drop of 50 µg/mL fibronectin (FN; Sigma-Aldrich) was placed on the PLL-PEG-coated surface of the coverslip and incubated for 1 hour. After 1 hour, the FN was removed, and the coverslip was washed once with distilled water. At this stage, the coverslip was ready to be printed with patterned FN on the PA gel surface. A thin film of FN was deposited on the areas of the coverslips that had been exposed to UVO light, whereas PLL-PEG prevented FN adhesion to the glass surface in the shadowed regions [64].

The polyacrylamide gels were fabricated with a mixture of acrylamide (Alfa Aesar) and bis-acrylamide (Fisher Bioreagent) following a established methods [65]. For traction force microscopy, fluorescent 0.2 µm diameter beads were added to the mixture for later use as fiduciary markers of gel deformation. Phosphate-buffered saline (PBS, Gibco) was used instead of distilled water to promote bead distribution toward the surface of the gel. The Young's modulus was controlled via the amount ratio of acrylamide/bis-acrylamide, as previously described [65]. Once both coverslips were ready, freshly made 10% ammonium sulfate (Sigma) and tetramethyl ethylenediamine (Sigma) were added to the polyacrylamide and bis-acrylamide mixture to initiate gel polymerization. Immediately after, a 2.5 µL drop of the mixture was pipetted on the treated glass bottom dish and sandwiched with the FN-patterned surface of the 12 mm round coverslip. The assembly was polymerized for 45 min before removing the round coverslip. The unpolymerized acrylamide was removed by rinsing twice with PBS. The resulting patterned gels were sterilized under 354 nm light for 5 min before adding the cells. The cells were seeded on top of the gels and allowed to adhere for 30 min. Unattached cells were washed off to avoid overgrowth of the patterns. The medium was reconstituted, and the cells were incubated overnight in the patterned regions.

## 3. Results

This section presents a computational verification and experimental demonstration of the proposed PIV-UQ and TFM-UQ techniques. PIV-UQ is verified using synthetic fluorescent bead images with prescribed displacement fields and noise sources in the image. PIV-UQ is shown to estimate substrate deformation uncertainty for each interrogation window. This information can be factored into the criteria to choose the critical PIV parameter, i.e., the interrogation window size, $W_L$. The synthetic simulations are then extended to the entire TFM pipeline to probe the influence of heteroskedasticity in PIV displacement data during the elastostatic inversion step. TFM-UQ applied to experimental data demonstrates the utility of the marginal posterior distribution in quantifying the variability due to hyper-parameter selection and numerical implementation, and the automatic identification of high traction stress error areas associated with image artifacts.

### 3.1. PIV-UQ identifies local uncertainty arising from image features and substrate deformation

The bootstrap PIV-UQ method was verified on a statistical ensemble of synthetically generated fluorescent bead images with spatially varying uncertainty. Experimental TFM bead images can be viewed as a realization of a complex image generation process and hence, the ability of PIV-UQ was tested using synthetic realizations containing additional image pixel noise. Fig A in S1 Text describes the computational pipeline to generate synthetic images for validation in detail. Briefly, 1024 × 1024 px images containing randomly placed synthetic beads with Gaussian intensity profiles (max. intensity 1000 a.u.) were generated. Each bead's

centroid was displaced according to a prescribed substrate deformation field, and different fields including translation and shear were simulated. Gaussian pixel noise was added to the image with varying levels of signal-to-noise ratio (SNR), as shown in Fig 5A and 5D. SNR is defined as $20\log(1000/\sigma_{px})$, where $\sigma_{px}$ is the std. dev. of the added pixel noise to the Gaussian bead image, saturated at an intensity of 1000 a.u. The process of random bead placement, displacement, and noise addition was repeated to generate ensembles of images. The RMS value of PIV-UQ uncertainty estimate of all such realizations ($\sigma_{u,\mathrm{PIV}}$) is compared with the ensemble uncertainty of baseline PIV measurement ($\sigma_{u,ens}$) over synthetic realizations (Fig A in S1 Text, total uncertainty, $\sigma = \sqrt{\sigma_{ux}^2 + \sigma_{uy}^2}$).

Two displacement fields were simulated (units in pixels):

$$\boldsymbol{u_1} = \begin{bmatrix} 2 \\ 0.6 \end{bmatrix}, \quad \boldsymbol{u_2} = \begin{bmatrix} \sin(\alpha) & g\cos^2(\alpha) \\ \cos(\alpha) & g\sin^2(\alpha) \end{bmatrix} \begin{bmatrix} x_0 \\ y_0 \end{bmatrix}. \tag{13}$$

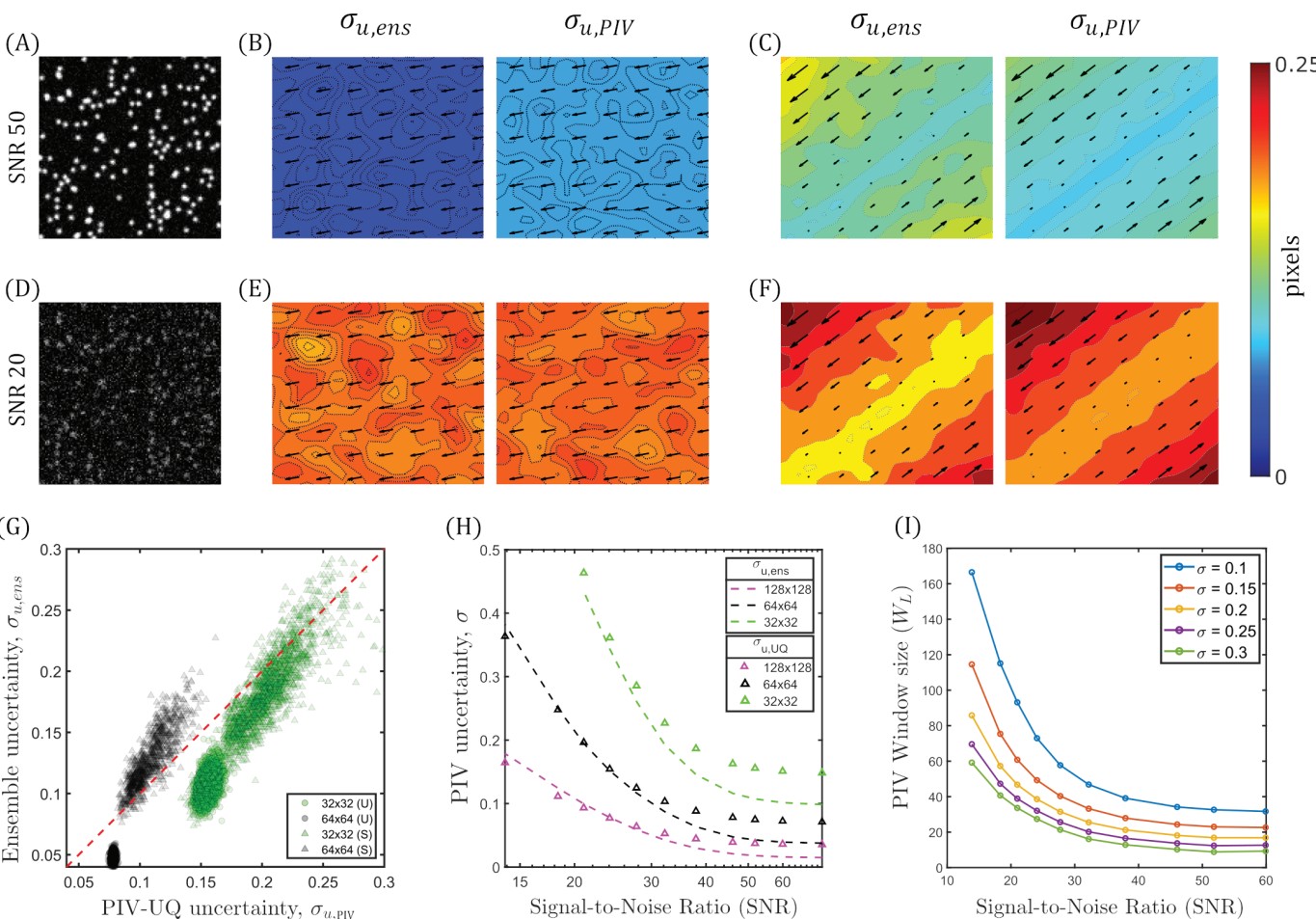

**Fig 5. PIV-UQ synthetic validation.** (**A**, **D**) Gaussian profile synthetic beads are randomly generated at varying signal-to-noise ratio (SNR) of image pixel noise. A uniform (U) displacement field (**B,E**) or a shear (S) displacement field (**C,F**) is applied to the synthetic beads and PIV-UQ is performed as described in §2.2. In (B,C,E and F), left columns show ground-truth data consisting of applied displacement field vectors overlaid on contour maps of ensemble standard deviation ($\boldsymbol{u}_{ens}$ and $\sigma_{u,ens}$). The right columns represent the same data estimated from PIV-UQ, $\boldsymbol{u}_{PIV}$ and $\sigma_{u,PIV}$. Units are in pixels (px). (G) Joint distribution of ($\sigma_{u,ens}$, $\sigma_{u,PIV}$) for uniform (U) and shear (S) deformation fields for $W_L$ of 32 and 64 px. Analysis procedure is described in Fig A in S1 Text (H) *RMS* of ensemble standard deviation and PIV UQ bootstrap estimates are compared for varying SNR values. (I) PIV Window size as a function of SNR with $\sigma_{u,\mathrm{PIV}}$ isolines derived from data presented in (H). N = 50 realizations of $1024 \times 1024$ pixels images were generated.

Here, $\alpha = 45°$, $g = 20$, and $x_0, y_0 \in [-1, 1]$ px for the simulation domain. In line with literature [47,51], a uniform displacement field $\boldsymbol{u_1}$ was simulated to understand the statistical convergence of the PIV-UQ bootstrap method for varying levels of image noise, quantified by the signal-to-noise ratio SNR. In addition, a second displacement field, $\boldsymbol{u_2}$, representing shear was used as a benchmark to mimic the substrate deformation gradients typically seen in TFM experiments.

The left panels of Fig 5B and 5C display arrow plots of the ground-truth, ensemble-averaged displacement fields, $\boldsymbol{u}_{ens}$ overlaid on contour plots depicting the ensemble standard deviation, $\sigma_{u,ens}$. The right panels show the displacements and standard deviation obtained by the PIV-UQ bootstrap method, $\boldsymbol{u}_{\text{PIV}}$ and $\sigma_{u,\text{PIV}}$. For the uniform displacement field ($\boldsymbol{u_1}$), $\sigma_{u,ens}$ was spatially uniform (Fig 5B). On the other hand, the shear displacement field ($\boldsymbol{u_2}$) showed a clear spatial gradient and reached higher values in the upper left and lower right corners of the image. Visual comparison of the ground truth and PIV-UQ data suggests that the bootstrap method was able to estimate PIV uncertainty and its spatial variations.

The uncertainty estimation of PIV-UQ became more accurate when the image SNR decreased (Fig 5D–F). To analyze in more detail the ability of PIV-UQ to distinguish regions of varying deformation uncertainty, we plotted scatter plots of *true* standard deviation of each displacement vector vs. the corresponding PIV-UQ estimate in Fig 5G (and Fig A in S1 Text). Uniform (U, circles) and shear (S, triangles) displacement fields are plotted using two different interrogation window lengths, $W_L = 32, 64$. The resulting point clouds approached the identity line $\sigma_{u,\text{PIV}} = \sigma_{u,ens}$ as uncertainty increased, but there was a non-zero $\sigma_{u,\text{PIV}}$ for zero true uncertainty, corresponding to the "floor" error level introduced by the bootstrap procedure.

Next, we analyzed the overall PIV uncertainty for the $\boldsymbol{u_1}$ field vs. *SNR* for different values of $W_L$ (Fig 5H). As expected, $\sigma_{\boldsymbol{u},\text{PIV}}$ decreased as SNR increased, and this dependence was steeper for smaller interrogation windows. In particular, $\sigma_{\boldsymbol{u},\text{PIV}}$ followed the ground-truth values closely for $SNR \lesssim 30$, but it saturated for higher SNR values to the floor noise level described above. Of note, Fig 5H illustrates the common knowledge that PIV error decreases as the interrogation window is enlarged, setting a trade-off between spatial resolution and measurement error. Since PIV-UQ can quantify this dependence, it can be practical to plot $W_L$ versus SNR for constant levels of $\sigma$, as in Fig 5I. The different curves in that plot indicate how the interrogation window size should be adjusted to keep a desired precision when performing PIV on images of varying quality. This approach offers an objective criterion for choosing $W_L$ in TFM experiments.

## 3.2. Heteroskedasticity of PIV measurement improves TFM regularization

Armed with the ability to estimate the spatially non-uniform (i.e., heteroskedastic) uncertainty in substrate deformation, we investigated how TFM-UQ propagated this uncertainty forward to the recovered traction stresses and how the regularization process was affected by heteroskedasticity. To establish ground-truth deformation data for this analysis, we used a synthetic traction stress field, $\boldsymbol{t}$, consisting of Gaussian traction islands ($\boldsymbol{t}_G$) i.e.,

$$\boldsymbol{t}_G = (t_{x,G}, t_{y,G}) = \left( t_p \exp\left\{ -\left[ (x - x_0)^2 + (y - y_0)^2 \right] / \left( 2s^2 \right) \right\}, 0 \right), \tag{14}$$

centered at $(x_0, y_0)$ with a width $s = 15\,\mu\text{m}$, and peak traction stress $t_p = 500\,\text{Pa}$. The stress field had four quadrants, each containing one traction island $\boldsymbol{t}_G$ pointing towards the vertical axis $x = 0$, as depicted in Fig 6A. The corresponding deformation field, determined by solving equation (1) using Lin et al.'s solution [58], is shown in Fig 6B. Synthetic fluorescent

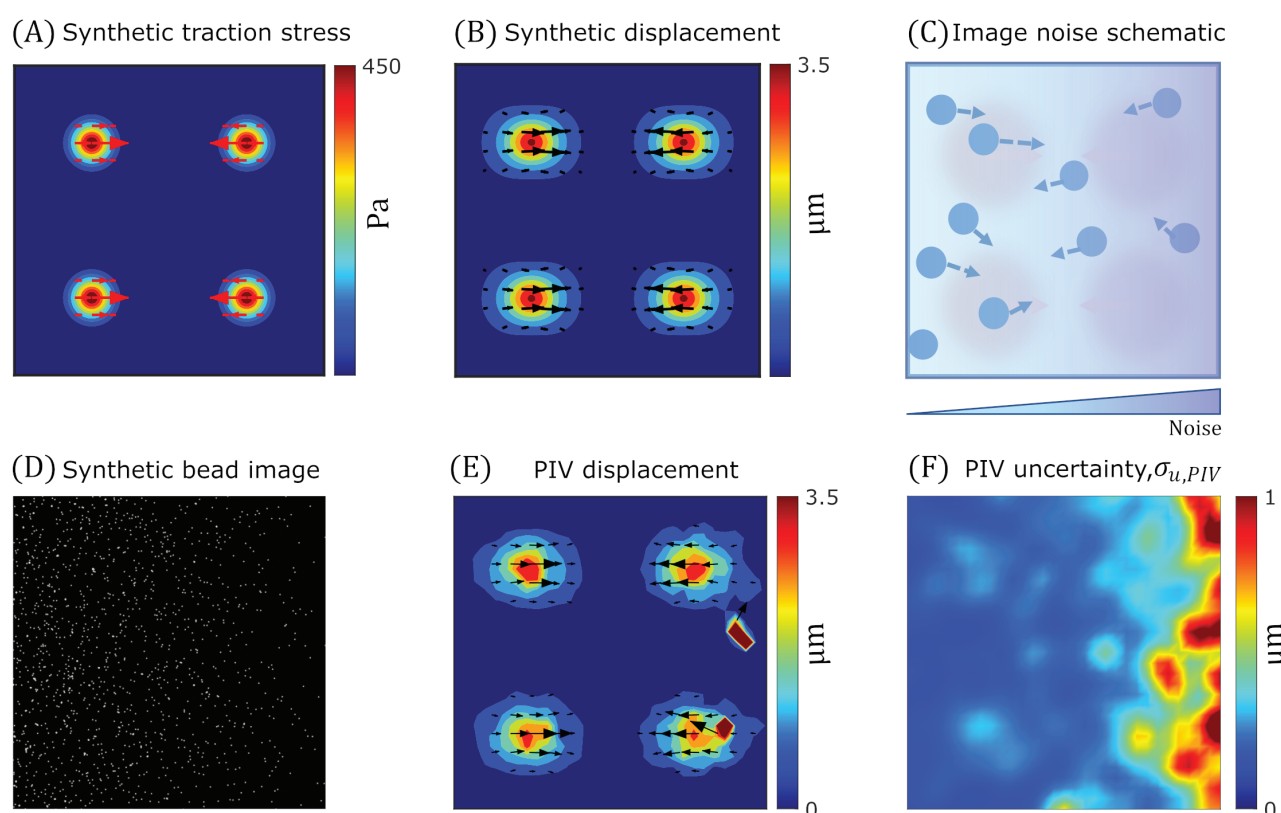

**Fig 6. Synthetic pipeline to simulate experimentally relevant, spatially heterogeneous noise levels in TFM.** (**A**) Synthetic traction stress generated from four traction islands of Gaussian profile, $t_G$, one in each quadrant with a maximum stress of 500 Pa. (**B**) Synthetic displacement field resulting from the traction profile in (A). Substrate of $E = 5$ KPa and Poisson's ratio of 0.45 is used here. (**C**) Schematic depicting spatially heterogeneous noise addition directly to the synthetic image by varying fluorescent bead density in X-direction. The synthetic image also has image pixel noise equivalent to SNR = 50 (**D**) Synthetic fluorescent bead image encoding image pixel noise and bead density variations. (**E**) PIV-UQ deformation measurement $\boldsymbol{u}_{\mathrm{PIV}}$ of the simulated bead images (D). (**F**) PIV-UQ estimation of uncertainty (standard deviation $\sigma_{u,\mathrm{PIV}}$) showing higher uncertainty increasing with x-direction in agreement with noise addition process (C).

bead images were randomly generated and displaced according to the computed deformation field, as described in §3.1. To consider heteroskedasticity, a horizontal image noise gradient was generated by manipulating the bead density (Fig 6C and 6D) while keeping a spatially constant pixel noise (SNR = 50). The resulting images were used as inputs to PIV-UQ to determine the displacements and the corresponding standard deviation fields, $\boldsymbol{u}_{\mathrm{PIV}}(x,y)$ and $\sigma_{u,\mathrm{PIV}}(x,y)$, are represented in Fig 6E and 6F. Consistent with the results shown in §3.1, PIV-UQ captured the horizontal gradient of deformation uncertainty, as well localized increases in $\sigma_{\boldsymbol{u},\mathrm{PIV}}(x,y)$ at the four traction islands, where deformation gradients were more intense. The PIV-UQ outputs were used to build

$$
\begin{aligned}
\boldsymbol{u}_{\mathrm{h}} = \{ & u_{PIV}(x_1,y_1),\dots,u_{PIV}(x_{Nw},y_{Nw}), \\
& v_{PIV}(x_1,y_1),\dots,v_{PIV}(x_{Nw},y_{Nw}) \}
\end{aligned}
\tag{15}
$$

and the covariance matrix

$$
\begin{aligned}
\boldsymbol{\Sigma}_{\mathrm{PIV}} = \mathrm{diag} \big\{ & \sigma_{u,\mathrm{PIV}}^2(x_1,y_1),\dots,\sigma_{u,\mathrm{PIV}}^2(x_{Nw},y_{Nw}), \\
& \sigma_{v,\mathrm{PIV}}^2(x_{Nw},y_{Nw}),\dots,\sigma_{v,\mathrm{PIV}}^2(x_{Nw},y_{Nw}) \big\},
\end{aligned}
\tag{16}
$$

where $N_w$ is the total number of interrogation windows. These variables were passed as inputs to the TFM-UQ framework described in §2.4.

The traction stress field recovered by TFM-UQ accurately approximated the prescribed ground truth $t$ (Fig 7A). It is worth pointing out that TFM-UQ adapted its regularizer to the spatially varying noise of the displacement field, delivering a uniformly smooth traction stress map despite the spatial noise gradient. Consistent with larger noise on the right-hand side of

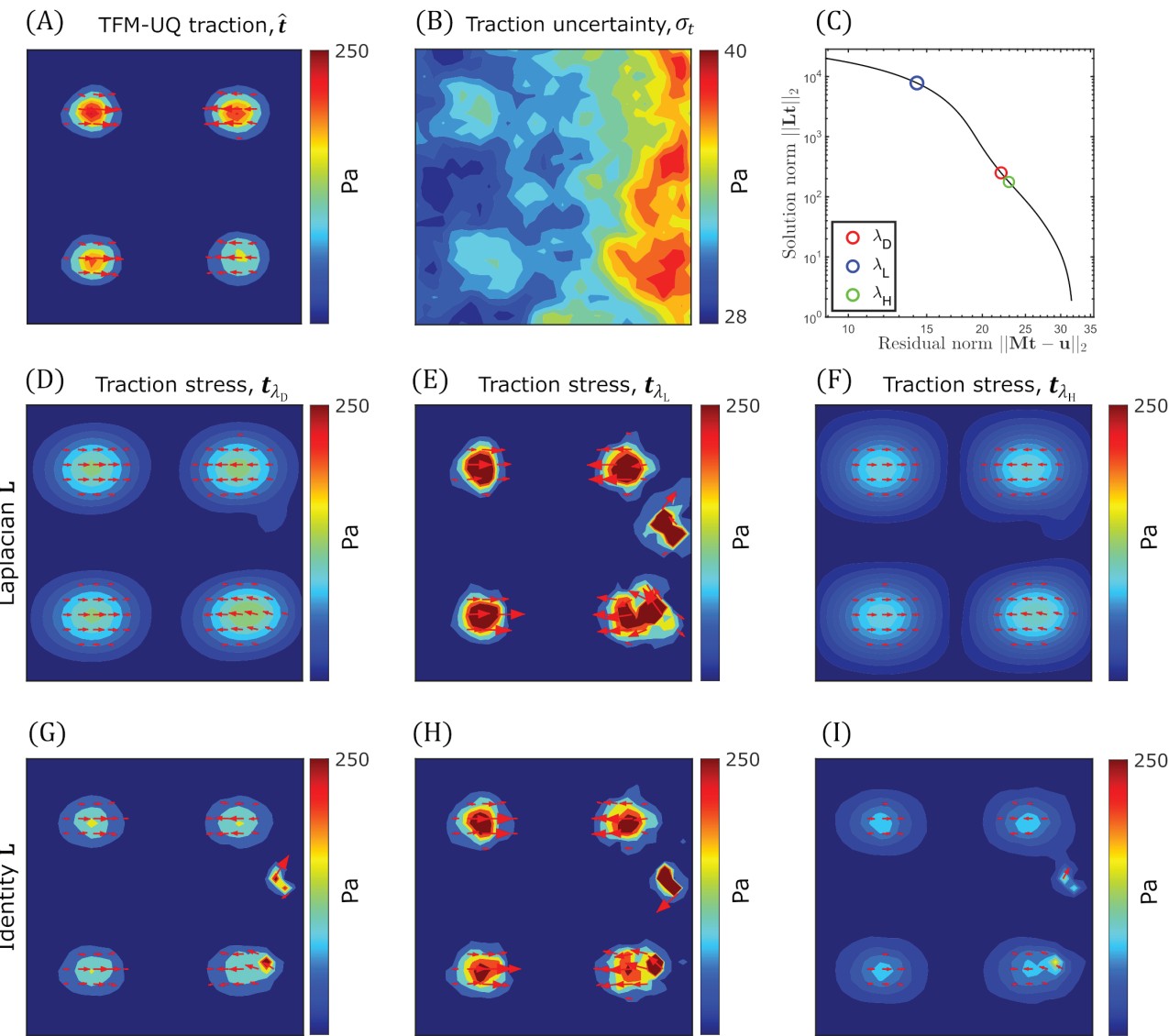

**Fig 7. TFM-UQ adaptively regularizes based on the local displacement uncertainty.** (A) Mean marginal posterior traction stress distribution ($\hat{t} = \mathbb{E}[t|u_h]$) approximated from Hybrid-Gibbs sampling. (Based on synthetic data described in Fig 6A.) (B) Pointwise marginal posterior uncertainty (standard deviation $\sigma_t$) of posterior traction stress (C) L-curve to determine Tikhonov regularization parameter $\lambda$ in traditional TFM methods. $\lambda_L$ and $\lambda_H$ are regularization parameters obtained from L-curve corners of homoskedastic synthetic simulations with spatially uniform low noise (L) or high noise (H). (D,E,F) Tikhonov regularized traction stress field corresponding to L-curve corner $\lambda_D$, low noise $\lambda_L$ and high noise $\lambda_H$ respectively. **L** is the discretized Laplacian operator. (G,H,I) Tikhonov regularized traction stress field, with **L** = **U** identity matrix. $\lambda_D, \lambda_L$ and $\lambda_H$ were determined to match $95^{th}$ percentile of traction stress magnitude with the respective fields in (D-F).

the image, regularization was stronger on that side. Moreover, TFM-UQ quantified the uncertainty of the recovered traction stresses at each point, defined as $\sigma_t = \sqrt{\sigma_{tx}^2 + \sigma_{ty}^2}$, (Fig 7B). This uncertainty increased from left to right, consistent with the noise gradient introduced in the image and the resulting $\boldsymbol{\Sigma}_{\text{PIV}}$ distribution (Fig 6F). It also showed local maxima at the traction islands, where deformation gradients were highest.

Despite exhibiting a spatial gradient, the left-to-right growth of $\sigma_t$ saturated quickly and its maximum value, $\sigma_{t,max} = 40$ Pa, was only 43% than its minimum value, $\sigma_{t,min} = 28$ Pa (Fig 7B). This uncertainty saturation was caused by the $\alpha$-dependent noise reduction prior in eq. 7, as expected from the Bayesian theorem. To illustrate this phenomenon, here we provide an estimate for $\sigma_{t|\alpha,\beta}$ using the diagonal of the posterior covariance matrix,

$$\mathbf{S}_{\text{post}} = \left[ \mathbf{M}^T \left( \beta^{-1} \mathbf{U} + \boldsymbol{\Sigma}_{\text{PIV}} \right)^{-1} \mathbf{M} + \alpha \mathbf{L} \right]^{-1}.$$

This matrix has a heteroskedastic contribution from the PIV measurement variance, $\boldsymbol{\Sigma}_{\text{PIV}}$ and a homoskedastic contribution from the $\beta$-proportional global error. Both contributions are affected by the elastostatic operator $\mathbf{M}$. In addition, $\mathbf{S}_{\text{post}}$ has a homoskedastic contribution from the $\alpha$-proportional noise reduction prior. Consider the model with identity prior (i.e., $\boldsymbol{\Sigma}_{\text{PIV}} = \mathbf{0}$ and $\mathbf{L} = \mathbf{U}$ [21,45]) equivalent to traditional Tikhonov regularization ($\lambda = \alpha/\beta$). In the long wavelength limit $kh \ll 1$, the elastic response becomes $\mathbf{M} \to h/\mu\mathbf{U}$, where $\mu = E/2(1 + \nu)$ is the shear modulus of the substrate [58]. In this case, $\mathbf{S}_{\text{post}}$ becomes diagonal with $\mathbf{S}_{\text{post},ii} \to \beta^{-1}[h^2/\mu^2 + \lambda]^{-1}$, bound between $\mu^2/h^2\beta^{-1}$ ($h^2/\mu^2 \gg \lambda$ limit, weak regularization) and $\alpha^{-1}$ ($h^2/\mu^2 \ll \lambda$ limit, strong regularization). If we allow for $\boldsymbol{\Sigma}_{\text{PIV}} \neq \mathbf{0}$ but make $\beta^{-1} = 0$ (neglecting $\boldsymbol{\varepsilon}_\beta$), the corresponding bounds for $\mathbf{S}_{\text{post},ii}$ are $\mu^2/h^2\boldsymbol{\Sigma}_{\text{PIV},ii}$ and $\alpha^{-1}$ for weak and strong regularization.

Despite $\boldsymbol{\varepsilon}_m$ inducing heteroskedasticity in the posterior traction stress distribution, the choice of a homoskedastic prior (i.e., global $\alpha$) sets the global scale bringing the posterior variance in $\mathbf{S}_{\text{post}}$ close to $\alpha^{-1}$. For a Laplacian regularizer $\mathbf{L}$ matrix, similar results were obtained (Fig 6A and 6B). The Bayesian framework provides a systematic way to understand the effect of prior construction on both the scale of recovered traction stresses as well as the uncertainty propagation. This analysis suggests that traction uncertainty is bound at each point by the largest contribution between regularization (global) and measurement noise (local), explaining why it does not fall to zero in areas of zero noise image. To palliate this floor effect, one could devise a heteroskedastic noise reduction priors such as replacing eq. 7 with

$$p(t|\alpha, \mathbf{L}, \boldsymbol{\Sigma}_{\text{PIV}}) \propto \exp\left[-\frac{\alpha}{2} t^T \boldsymbol{\Sigma}_{\text{PIV}}^{-1} \mathbf{L} t\right],$$

which would make both the minimum and maximum values of the diagonal of $\mathbf{S}_{\text{post}}$ proportional to $\boldsymbol{\Sigma}_{\text{PIV}}$ under the assumptions outlined above. However, this prior would also provide weaker regularization in high-noise areas. Repeating the analysis in Fig 7A and 7B for this heteroskedastic noise reduction prior confirmed these predictions (see Fig B in S1 Text, bottom row).

To further understand how TFM-UQ incorporated heteroskedasticity to regularize traction stresses, we compared this method's results with those obtained using a conventional, homoskedastic Tikhonov regularizer based on the same Laplacian kernel $\mathbf{L}$ and the elastostatic operator $\mathbf{M}$. The conventional regularizer depends on one hyperparameter, $\lambda = \alpha/\beta$, which was chosen by analyzing the L-curve, i.e., a log-log plot of $|t^T \mathbf{L} t|$ versus $|(\boldsymbol{u}_{\text{h}} - \mathbf{M}t)^T (\boldsymbol{u}_{\text{h}} - \mathbf{M}t)|$ (Fig 7C), to balance the solution's smoothness and its fit to the data as described previously [3,46]. The value of $\lambda$ at the point where the L-curve has the maximum curvature

(the corner of the L-curve) is usually selected to regularize $t$. Higher values of $\lambda$ underfit the traction stresses and are located to the right of the L-curve corner. Conversely, lower $\lambda$ values overfit and are located to the right of the corner. This standard approach attempts to find a one-fits-all value of $\lambda$ for the whole image. This can lead to underfitting and overfitting in different regions of experiments with spatially dependent noise. Here, it is important to remark that the noise not only comes from the image, but also from the deformation imparted by the cells in the substrate, so local gradients of noise can arise even in high-quality images with uniform bead distributions.

In our validation dataset, we tested the conventional regularizer for three $\lambda$ values: the optimal value $\lambda_D$ corresponding to the L-curve corner, as well as a pair of values $\lambda_L(<\lambda_D)$ and $\lambda_H(>\lambda_D)$ located on opposite sides of the corner. The $\lambda_L$ and $\lambda_H$ values were selected as the L-curve corner for synthetic simulations with low and high levels of spatially uniform noise (Fig C in S1 Text, left and right columns respectively).

This corresponds to the left and right edges of the differential noise image (Fig 6C and 6D). Compared with the ground-truth traction stresses shown in Fig 6A, the **L**-based Tikhonov regularizer underestimated and oversmoothed the traction stress field for $\lambda = \lambda_D$ (Fig 7D), showing significantly more spread patterns and lower traction stress values than TFM-UQ, without completely cancelling the noise (see the defect in the upper right quadrant). For $\lambda = \lambda_L$, **L**-based Tikhonov did not underestimate traction stresses values or spread their pattern (Fig 7E). However, this method produced significant noise and artifacts, including spurious concentrations of considerable traction stress in areas with zero ground-truth values. Finally, **L**-based Tikhonov severely underestimated and oversmoothed the traction stresses for $\lambda = \lambda_L$, producing non-zero values of $t$ almost everywhere in the region of interest (Fig 7F). This effect of oversmoothing in regions of high noise is consistent with the behavior of L-curve based Tikhonov regularizer in simulations of spatially uniform with a low bead density (Fig C in S1 Text, middle right). Traditional regularization is unaffected for images with low noise and high information content (Fig C in S1 Text, middle left). On the other hand, TFM-UQ avoids underfitting in low density (high noise) while qualitatively similar results with images of low noise (high density) (Fig C in S1 Text, bottom).

For reference, we also recovered traction stress maps for $\lambda_D$, $\lambda_L$, and $\lambda_H$ using a Tikhonov regularizer with the identity matrix (i.e., **L** = **U**) instead of the discretized Laplacian operator since this regularizer was preferred in several previous studies [21,23]. The values of $\lambda$s were selected so that the $95^{th}$ percentile of the traction stress field magnitude with **L** = **U** matches with their counterparts using the Laplacian **L** precision matrix. These maps, displayed in Fig 7G–7I, showed a similar trend as those obtained with the Laplacian smoother in their variation with $\lambda$. They had less spread stress contours but displayed more noise and spurious traction islands.

It is also worth noting that, while they spread out stresses in space, Laplacian regularizers conserve the total traction forces (area integral of traction stress). Noting that the Laplacian is the divergence of the gradient, this property is an immediate corollary of Gauss's theorem for any distribution of traction stresses applied within a bound region. Conversely, non-Laplacian Tikhonov regularizers do not necessarily conserve traction forces.

The behaviors visualized in Fig 7 manifested quantitatively in metrics of the recovered traction stress fields, summarized in Table 1. TFM-UQ provided the minimum root mean square (RMS) error with respect to the ground truth, and this error was well approximated by the TFM-UQ RMS $\sigma_t$ estimate. To quantify how different regularizers balanced overfitting and underfitting, we defined the foreground traction map as the region where the true stress magnitude $|t| \geq \text{RMS}(t)$ (i.e., 75 Pa for the synthetic field of 6 A). Likewise, the background traction map was defined as the region where $|t| < \text{RMS}(t)$. We then evaluated the

**Table 1. Performance comparison of locally regularized TFM-UQ with classical (global) Tikhonov regularization (discrete Laplacian Prior).** The **best** and *worst* performing methods are highlighted in bold and italic respectively for each metric.

|  | True | TFM-UQ | $\lambda_D$ | $\lambda_L$ | $\lambda_H$ |
|---|---|---|---|---|---|
| Weighted RMS error in $t$ (Pa) | - | **26.44** | 38.48 | *64.46* | 41.51 |
| RMS $\sigma_t$ (Pa) | - | **23.36** | N/A | N/A | N/A |
| Foreground mean $t$ (Pa) | 182.22 | 99.94 | 77.78 | **170.13** | *63.53* |
| Foreground max $t$ (Pa) | 496 | 252.95 | 138.57 | **477.31** | *110.17* |
| Background mean $t$ (Pa) | 8.65 | **14.36** | 23.67 | *66.47* | 22.38 |
| Background max $t$ (Pa) | 76.79 | **88.55** | 119.20 | *1428.77* | 96.25 |
| $SNR_t$ | 21.05 | **6.96** | 3.28 | *2.56* | 2.84 |

mean and maximum traction stress magnitude of the foreground and background maps. A TFM method with a good overfitting-underfitting balance should yield both a large foreground traction magnitude and a low background traction magnitude. This analysis, also summarized in Table 1, showed that TFM-UQ outperformed the standard regularizers in all metrics except in mapping the foreground traction, where it ranked second best after the standard $\lambda_L$ regularizer. However, $\lambda_H$ performed worst among all methods in RMS error, keeping a low background traction and overall SNR of the traction map. Overall, TFM-UQ showed the best balance between overfitting and underfitting by considering the local deformation uncertainty from PIV-UQ. In addition, TFM-UQ was able to propagate PIV uncertainty to estimate the error of the recovered traction stresses.

## 3.3. Experimental demonstration of PIV-UQ and TFM-UQ

This section describes two sets of live cell experiments demonstrating the practical application of PIV-UQ and TFM-UQ, and illustrate how these methods deal with spatially varying error in the measured substrate deformation. First, we performed experiments on C3H/10T1/2 fibroblasts seeded on polyacrylamide substrates (Young's modulus of 5 kPa) with micropatterned square fibronectin islands of width 75 μm (Fig 8A). Micropatterning extracellular matrix proteins on cellular substrates is a common approach for guiding adherent cells to adopt predefined shapes [64]. In our study, we used patterned substrates as a benchmark for TFM-UQ, because these substrates can exhibit gradients of fluorescent bead brightness and background noise, as shown in Fig 8B. In addition to the spatially heterogeneous bead noise, we compared the traction stresses recovered from the "raw" images obtained by wide-field fluorescence microscopy as well as from a second dataset (Fig 8C) obtained by applying a non-linear background subtraction ("BGS") method (Leica Instantaneous Image Clearing [66]) that improved the signal-to-noise of the image by a factor of 6. This feature allowed us to objectively assess PIV-UQ and TFM-UQ against different levels of spatial image noise when observing the same cell at the same instant of time and subjected to different level of image processing.

Fig 8D and 8G shows the in-plane substrate deformations measured by PIV-UQ for these two datasets using PIV interrogation windows of length $W_L = 128$ pixels, $\boldsymbol{u}_{\text{PIV},128}(x, y)$. These vector maps were overlaid on their associated uncertainty maps, $\sigma_{\boldsymbol{u},\text{PIV},128}(x, y)$. Consistent with the synthetic analyses presented in §3.1, the PIV uncertainty was markedly higher in the raw image. Also consistent with the synthetic analyses, regions with deformation gradients such as the edges and corners of the islands, also exhibited higher PIV uncertainty. Coefficient of variation (CoV) was used to determine convergence of the estimator $\sigma_{u,\text{PIV}}$. Fig D

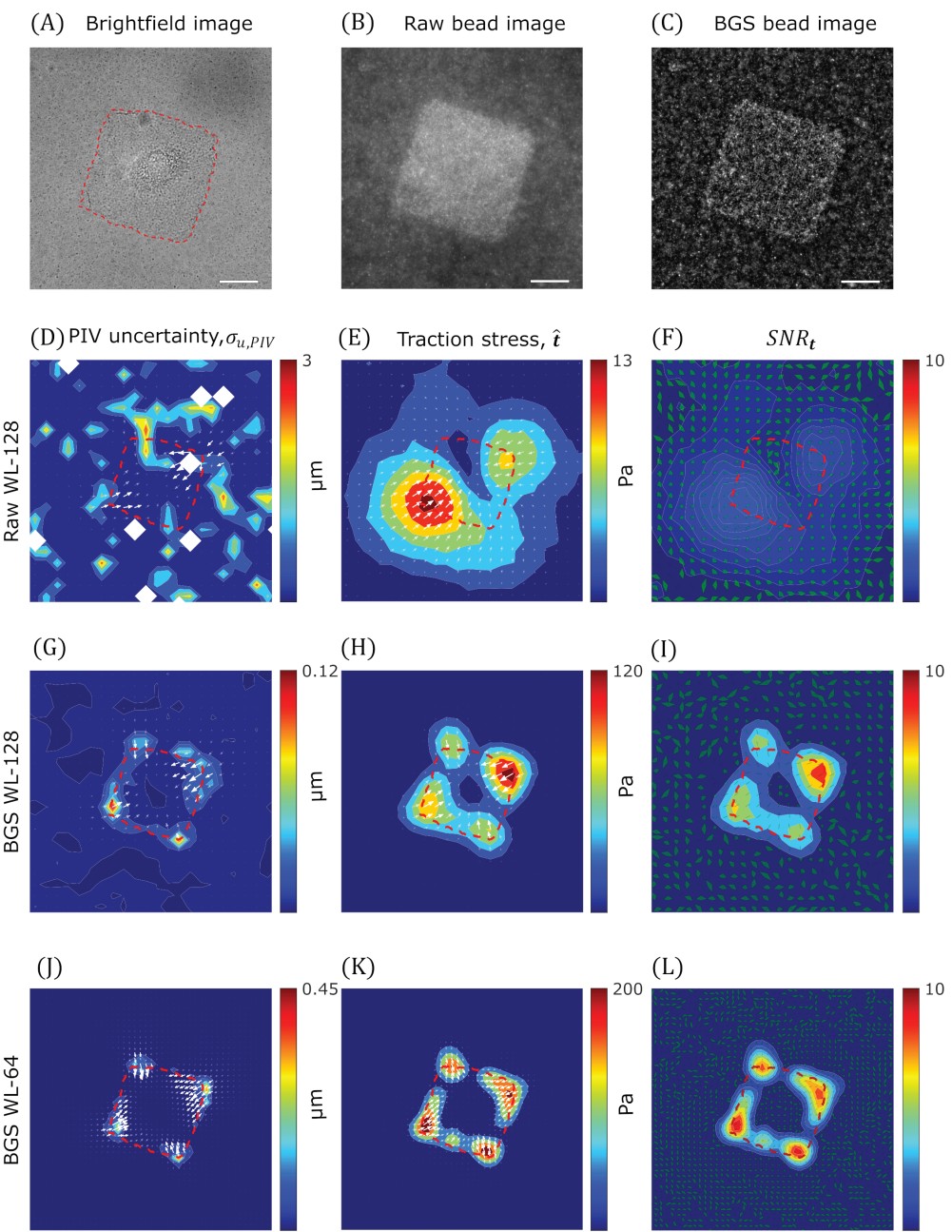

**Fig 8. TFM-UQ captures variability associated with microscopy image quality.** (**A**) Brightfield image of C3H/10T1/2 cell cultured on fibronectin micropatterned island of length 75 μm (Red dashed outline). (**B**) "Raw" wide-field fluorescent bead image (beads of size 0.2 μm). (**C**) "BGS" Background subtracted image processed from (B). (**D,G,J**) $u_{\text{PIV}}$ (arrows) overlaid on PIV-UQ uncertainty field ($\sigma_{u,\text{PIV}}$) corresponding to raw WL-128, BGS WL-128 and BGS WL-64 respectively. Here, WL denotes the PIV interrogation window size $W_L$. (**E,H,K**) Mean marginal posterior traction stress field ($\hat{\mathbf{t}}$) corresponding to Raw WL-128, BGS WL-128 and BGS WL-64 respectively. (**F,I,L**) Traction stress signal-to-noise ratio ($SNR_t$) plotted as a heatmap corresponding to Raw WL-128, BGS WL-128 and BGS WL-64 respectively. Overlaid uncertainty arrows denote the pointwise angular uncertainty corresponding to 1 circular std. dev. of marginal posterior $p(\mathbf{t}|\mathbf{u}_{\text{h}})$. Scale bar: 25 μm.

in S1 Text shows that this convergence depends on the resolution as well as the image quality (e.g., to achieve 10% variability, raw images required $n_B \approx 48$ iterations, while BGS images

only required $n_B \approx 7$ iterations). Fig 8E and 8H shows the traction stress maps recovered from the raw and background subtracted (BGS) images for $W_L = 128$ pixels. The traction stresses recovered from the raw images were markedly weaker and their spatial patterns were more spread out than those obtained from the background subtracted images. The differences were due to the more aggressive regularization applied to the PIV vectors from the raw images, which had significantly more uncertainty.

These data illustrate that the effective spatial resolution of the traction stresses recovered by TFM-UQ is not uniquely determined by the PIV resolution. While this is a relatively well-known consequence of regularization shared with other TFM methods [23,42], it is worth noting that TFM-UQ uses an objective criterion to select the level of regularization automatically. Moreover, the traction stress uncertainty, $\sigma_t$, can be used to quantify the signal-to-noise ratio in the recovered traction stresses, $SNR_{t,128} = |\hat{\mathbf{t}}_{128}|/\sigma_{t,128}$. Fig 8F and 8I displays these $SNR_{t,128}$ for raw and BGS images together with vector fans whose width represents the angular uncertainty in the orientation of the $t$ vector. The figures reveal that the largest variability in the direction of $t$ was observed at locations of low traction stress magnitude. Moreover, and as expected, these data indicate that the $SNR_{t,128}$ obtained from the raw images was significantly worse than that obtained from the background subtracted ones with higher contrast.

The results presented above verify that the $SNR_t$ maps obtained by TFM-UQ effectively capture how image quality affects traction stress recovery. A practical application of $SNR_t$ is to evaluate how varying $W_L$ affects this recovery. To illustrate this idea, we re-ran PIV-UQ and TFM-UQ for the BGS images at higher resolution, i.e., using $W_L = 64$ pixels, and asked if this change produced a more precise traction stress map. The answer to this question is not trivial and depends on whether the gain in PIV resolution (Fig 8J), which is accompanied by an increase in peak traction stresses at cell-substrate attachments (Fig 8K), compensates the increase in the PIV measurement error and its propagation to the traction stresses. In this case, comparing Fig 8I and 8L indicated that $SNR_{t,64} > SNR_{t,128}$ and, therefore, that refining the PIV window size led to a more precise representation of the traction stresses. Repeating the analysis for $W_L = 32$ pixels (Fig E in S1 Text) did not yield a higher $SNR_{t,32}$ everywhere, compared to $SNR_{t,64}$, showing a notable decrease in the top-right corner. Likewise, the traction stresses recovered for $W_L = 32$ did not display a significantly higher spatial resolution than those obtained for $W_L = 64$ as the regularization was more aggressive for the $W_L = 32$ dataset. This example illustrates how TFM-UQ can be applied to objectively choose the optimal PIV window size that balances resolution with error, which in this case would be $W_L = 64$ pixels.

In addition, TFM-UQ outputs posterior traction uncertainty maps that can be used to distinguish the TFM measurement noise from cell-to-cell biological variability. This notion is illustrated by comparing the maximum traction stress of C3H/10T1/2 fibroblasts seeded on micropatterned square fibronectin islands of two different sizes (Fig F in S1 Text). Significantly higher peak stresses are observed in the $75 - \mu m$ islands compared to the $50 - \mu m$ ones. Additionally, TFM-UQ indicates that the traction stresses on the $75 - \mu m$ islands show high variability among different cells, which cannot be explained by measurement noise alone. In contrast, the cell-to-cell variability is markedly lower in the $50 - \mu m$ islands, and the only two notable outliers have large measurement error bars. This example clearly shows how the uncertainty quantification in TFM-UQ can help interpret biological experiments. To verify that the posterior traction stress uncertainty is not just simply related to the traction magnitude, Fig F in S1 Text (bottom right) shows a scatter plot of these quantities, revealing lack of correlation.

A second set of experiments was performed with human umbilical vascular endothelial cells (HUVECs) seeded on fibronectin-coated PAA gels to form confluent monolayers (Fig 9A). To illustrate how TFM-UQ handles images with distinct areas of poor quality, we selected an example containing a large clump of fluorescent beads and an out-of-focus region, indicated by the arrows in Fig 9B. Fig 9C displays the substrate deformation map obtained by PIV-UQ for this experiment, showing significant spatial gradients consistent with previously reported data from HUVEC monolayers [33]. The associated PIV-UQ uncertainty maps in vector magnitude and direction, shown in Fig 9D, indicate that the PIV errors were mostly localized to regions of poor image quality and spatial gradients of $\boldsymbol{u}_{\mathrm{PIV}}$, in agreement with the results presented in §3.1. In the bead clump, PIV-UQ failed to provide deformation vectors leading to the empty white areas in Fig 9C–9D. To feed input to the TFM-UQ algorithm in this empty region, the displacement vectors were prescribed using a median filter, and $\boldsymbol{\Sigma}_{\mathrm{PIV}}$ was set to equal the maximum value from the rest of the image. TFM-UQ used the $\boldsymbol{u}_{\mathrm{PIV}}$ and $\boldsymbol{\Sigma}_{\mathrm{PIV}}$ fields to recover and regularize the traction stresses (Fig 9E), and mapped the uncertainty in the magnitude and direction of the stress vectors (Fig 9F). It is worth noting that the areas of highest $\sigma_t$ co-localized with the bead clump, the out-of-focus region in the fluorescent bead image and with the largest deformation gradients. These results illustrate how TFM-UQ

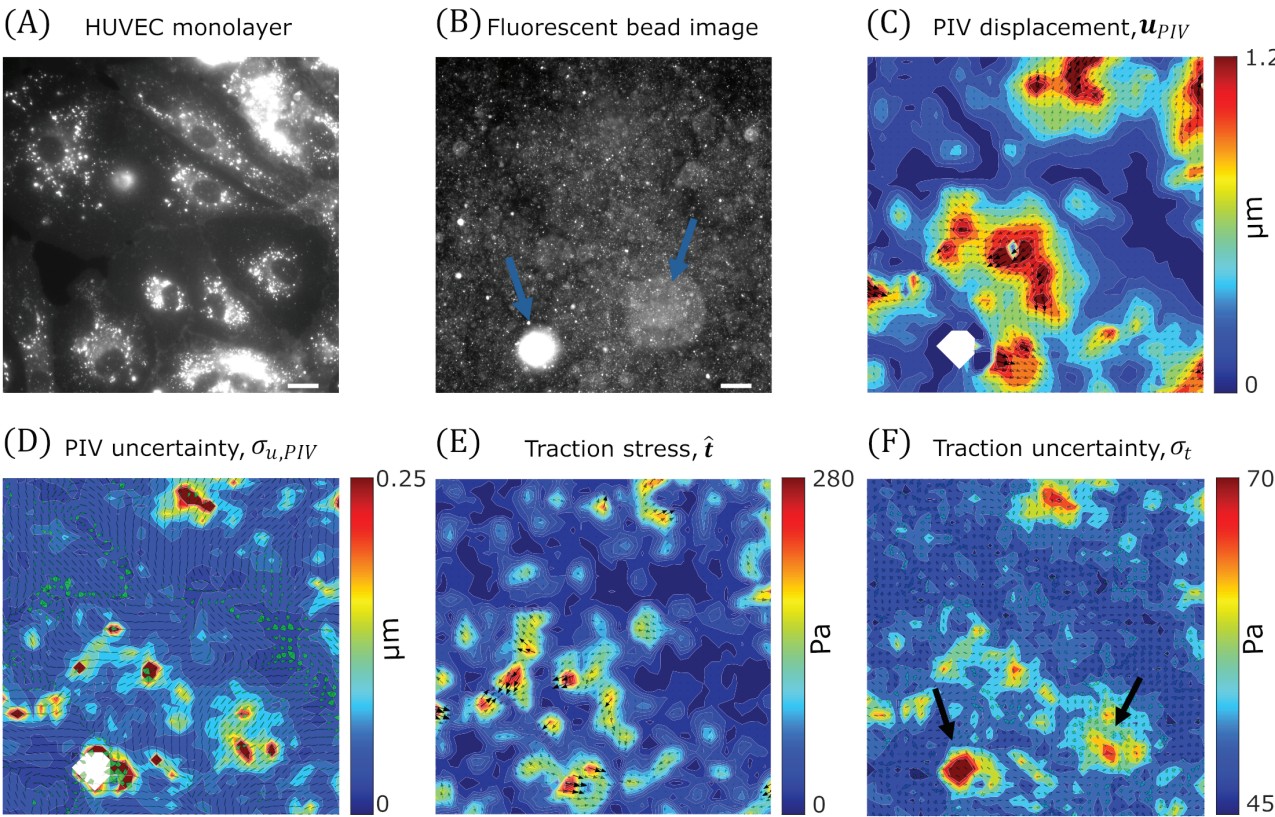

**Fig 9. TFM-UQ applied to endothelial monolayer experiment demonstrates uncertainty propagation.** (A) Membrane labeling of HUVEC monolayers with CellMask. (B) Corresponding fluorescent bead image (beads of size 0.2 μm). Arrows indicate bead-related image artifacts (C) PIV-UQ displacement field $\boldsymbol{u}_{\mathrm{PIV}}$ (D) PIV-UQ uncertainty map, $\sigma_{u,PIV}$. White regions indicate "bad" PIV windows that were deleted and replaced as described in §2.2. Uncertainty arrows denote the pointwise angular uncertainty corresponding to 1 circular std. dev. of bootstrapped PIV-UQ distribution. (E) Inferred mean marginal posterior traction stress, $\hat{\mathbf{t}}$ (F) Marginal posterior traction stress uncertainty field ($\sigma_t$). Uncertainty arrows denote the pointwise angular uncertainty corresponding to 1 circular std. dev. of marginal posterior $p(\boldsymbol{t}|\boldsymbol{u}_{\mathrm{h}})$. Scale bar : 25 μm

integrates information from PIV measurements and image quality to map the confidence of the recovered traction stresses.

## 4. Discussion

Since the pioneering silicone membrane wrinkling assay [1] demonstrated the possibility of detecting cell-generated traction stresses, various strategies have been exploited to quantify these stresses. Currently, a common approach involves measuring substrate deformation by tracking fiduciary markers (e.g., fluorescent beads) embedded in elastic substrates of known mechanical properties. Then, the traction stresses corresponding to the measured deformation field are recovered by solving an inverse elastostatic problem. The shortage of tools to quantify how measurement uncertainty propagates to the recovered stresses hampers the objective evaluation of experimental protocols and elastostatic inversion methods. These limitations make it impossible to separate biological variability from measurement error and promote the view that TFM technique is susceptible to methodological arbitrariness [42,52,67].

### 4.1. Uncertainty quantification in TFM

One of the most common methods to measure the cell-induced substrate deformations for TFM is particle image velocimetry (PIV), a technique originally developed to measure velocity vectors in fluid flows [55,68]. PIV measurement errors are associated with the signal-to-noise ratio (SNR) of the image and spatial deformation gradients. In a typical TFM experiment, the image SNR can be optimized by choosing fluorescent beads of the appropriate size, and controlling their density and spatial distribution during substrate fabrication. Image quality can also be tuned by adjusting substrate thickness, the optics and operational settings of the microscope used for acquisition, and post-processing of the acquired images [23,30,67,69]. Batch-to-batch variability in reagents and operator dependence in fabrication and imaging protocols can cause image quality variations (bead clumping, focus and bead density gradients, etc.) within and across experiments. In addition, differences in equipment, experimental, and analysis protocols among laboratories can create experimental uncertainty in TFM. Ultimately, PIV uncertainty is impossible to control *a priori* since it also depends on the deformation gradient generated by the cells on the substrate. Therefore, it is necessary to develop tools to estimate PIV uncertainty in individual TFM experiments and propagate this uncertainty to the recovered traction stresses.

The fluid mechanics community has devoted extensive research to quantifying PIV uncertainty [47–49], but significant differences between experimental images generated in elastic substrates and fluid flows impede the transfer of that knowledge to TFM. Typically, TFM images have limited bead brightness due to concerns over phototoxic effects. The substrate deformations are relatively small and experience gradients over micrometric lengthscales. Moreover, ensemble correlation approaches do not improve PIV resolution in TFM since the bead pattern does not vary over time, unlike in flowing fluids. In addition, TFM experiments can last hours and are more sensitive to drift and positioning errors along the optical axis. Consequently, image cross-correlation peaks are broader and more irregularly shaped in TFM than fluid flow experiments. This feature makes the parametric methods widely used to estimate uncertainty in fluid flow less well suited for TFM.

This work introduced a non-parametric bootstrap method to estimate PIV uncertainty in TFM experimental images (PIV-UQ). Bootstrap resampling is a powerful tool for constructing accuracy estimates without ground truth by perturbing limited data. Since the PIV estimate is exclusively based on the information contained within the interrogation windows

($\{\mathbf{W}_{ij}\}$), PIV-UQ teases out the possible displacement vectors by perturbing different pixel contributions to the final estimator. In PIV-UQ, the image pixels are bootstrapped multiple times to produce a distribution of measured substrate displacement vectors for each interrogation window, from which the most likely displacement and its standard deviation are calculated. Thus, the PIV-UQ estimates the variability or stability of PIV estimate due to the non-uniform motion of tracers, pixel noise due to experimental and imaging or post-processing employed. This approach generalizes Kybic's [51] bootstrap method by introducing perturbations directly in the image pixels instead of in the optimization that calculates the deformation. Therefore, the bootstrap method derived for PIV-UQ in this work could be applied to other image registration techniques, such as optical flow [43,70,71] and digital image correlation [72,73], also used in cellular force measurement. Apart from TFM experiments, PIV is routinely used in biomechanics and mechanobiology research to estimate *in vitro* and *in vivo* tissue flow [6,7,74], and cardiomyocyte contractility assays [30,75]. The PIV-UQ tools developed here should be directly transferable to those applications.

Bayesian methods model the quantities of interest as random variables, assimilating information regarding physical models, input data, prior constraints, and their variability. Thus, these methods offer a natural framework to propagate substrate deformation uncertainty to the traction stresses recovered by TFM. Bayesian methods were proposed in Dembo and Wang's seminal TFM work [9], and the interest in adapting them to TFM is recently increasing [21,43,76]. Traditionally, Bayesian TFM workflows have focused on elastostatic inversion without explicitly considering the uncertainty in substrate deformation or propagating this uncertainty. Interestingly, Butler *et al*'s pioneering constrained FTTC method [10] considered the possibility of nonphysical displacement vectors and involved an iterative algorithm to correct these measurements so that the traction stresses satisfied a prior constraint. A more detailed Bayesian TFM model including cell area as a prior constraint has been developed recently [76]. However, both of these methods do not propagate measurement errors from the images or the displacement field. To the best of our knowledge, the only effort to propagate uncertainty from the images to traction stresses is Boquet-Pujadas *et al*'s optical flow method for substrate deformation [43], where traction stresses are modeled in the image registration process. This approach considers the traction stresses the only cause for deviations in pixel intensities in the image set $\boldsymbol{I}$. While the spatial variability of error is incorporated in the framework, the work was primarily theoretical and did not consider experimental data. In particular, it would be interesting to assess how the framework handles image error sources that are not explicitly modeled (e.g., fluorescent bead artifacts, camera focus variation, etc).

In this work, we developed TFM-UQ, a hierarchical Bayesian framework that propagates the PIV measurement variability estimated by PIV-UQ and provides point-by-point estimations of traction stress uncertainty. The resulting uncertainty maps, $\sigma_t(x, y)$, establish error bars for the traction stress recovered at each location, both in magnitude and direction of the stress vector. This information can help interpret and analyze TFM experiments for multiple reasons. Among these, it may help identify outliers automatically and objectively. As demonstrated in §3.3, it allows for separating biological and measurement uncertainties when pooling data from repeat experiments. It can also help pinpoint sources of measurement error to refine experimental protocols and image analysis methods. Most importantly, the combination of traction and uncertainty maps (e.g. $SNR_t$) allows users to understand how TFM is balancing the weight of the image data with regularization levels. For images with spatially varying noise levels, TFM-UQ uncertainty maps reveal the local regions of low confidence. In this way, TFM-UQ distills the quality of image, the effect of PIV window and regularization

on the final output that informs the user of the measurement confidence in the results. In contrast, frequentist approaches can mask errors by introducing global over-smoothing, creating a false confidence in the results [77].

Applying PIV-UQ to fluorescent bead images from TFM experiments, we demonstrated this method quantifies how the PIV error depends on the image quality and the size $W_L$ of the interrogation window. The latter is a key PIV parameter affecting measurement error and resolution. TFM protocols often provide loose guidance suggesting $W_L$ should be small enough to resolve the relevant spatial features of cell-generated deformations and large enough to minimize PIV error [30,59]. The scarcity of UQ tools in TFM has hampered the development of more quantitative criteria. The PIV-UQ and TFM-UQ tools presented in this manuscript can be used independently and/or together to objectively choose the optimal PIV window size for each experiment. Using PIV-UQ alone without TFM-UQ, one could analyze one or a few fluorescent bead images selected from a batch of frames (e.g., frame of maximum contraction) in a time-lapse experiment to adjust $W_L$, so that the PIV measurement error remains within expectations and is conserved across experiments. Using PIV-UQ and TFM-UQ together, it is possible to identify the value of $W_L$ that maximizes the resolution and precision of the recovered traction stresses for each experiment.

## 4.2. Uncertainty-aware elastostatic inversion and regularization

A variety of mathematical models have been used to represent the elastostatic response of TFM substrates and an even wider variety of techniques have been proposed to invert the model equations, including boundary element methods [9,21,23,40], Fourier expansions providing exact analytical solutions in frequency space [10,11,23,58,59], finite element methods (FEM) [13,78–81], Bayesian methods [21,31,43,45,76], etc. Owing to the physics of elastostatics, this inversion amplifies the small-scale (high-frequency) features of the measured deformation field. Because measurement noise typically has a high content of small-scale features, TFM outputs can suffer from significant noise even when using exact analytical solutions for inversion. In addition, discretized numerical approximations of the elastostatic response can lead to singular or near-singular linear systems of equations, especially when the domain size and spatial resolution increase. Therefore, regularization is a crucial aspect of image analysis in TFM.

Regularization and data smoothing are routinely employed to enforce well-behaved traction stress fields and suppress the amplification of high-frequency noise during the inversion. For data with minimal high-frequency corruption, direct inversion can be achieved [10,77,79]. Many implementations perform implicit data filtering to stabilize the numerical inversion, e.g., frequency cut-off in Fourier based methods [10,23]. In FEM methods, nodal interpolation with low-order polynomials effectively filters out high frequencies [79,79,82]. In synthetic studies, an explicit regularization term in the cost function has been shown to recover traction with lower error than unregularized Fourier or FEM algorithms [23,77,81]. The specifics of such explicit regularization algorithm can vary among TFM formulations, but in general, all regularization schemes introduce a penalty term whose residual must be minimized together with the elastostatic residual. Regularization methods generally require specifying the values of the hyper-parameters controlling each residual's weight in the global minimization process. Heuristic approaches to hyperparameter selection, such as the L-curve criterion [3,46], have enjoyed popularity, but they are susceptible to significant user bias and are often challenging to interpret objectively [3,42,67]. Self-consistency [39,43,67] is an appealing principle to find regularization parameters, but it requires *a priori* knowledge of $\Sigma_{\text{PIV}}$, which

was not previously available. The PIV-UQ technique presented in this manuscript paves the way for a systematic application of this principle.

The interest in Bayesian methods to select hyperparameters has increased in the last decade [21,31,43,45]. These methods formulate hyperparameter inference as the maximization of the posterior joint probability $p(\lambda, t|u_h)$ or the evidence $p(\lambda|u_h)$, where $\lambda$ is model hyperparameter(s) and $u_h$ the measured substrate deformation. While this unified framework is attractive and has built-in self-consistency, the existing works pay little attention to the uncertainty in PIV-measured substrate deformation or the variability in hyperparameter selection. There is an abundance of methods (e.g., low-pass filtering, median filtering, etc) to "pre-regularize" the $u_h$ obtained from PIV or equivalent technique prior to traction stress recovery. However, once the measurements are pre-regularized, the information about which PIV vectors are less reliable than others is not passed on to the TFM algorithm. Consequently, existing Bayesian methods make strong assumptions about the nature of the noise distribution, i.e., that it is spatially uniform and independent of the effects of cellular forces [21,23]. These assumptions have not been verified systematically.

To overcome the drawbacks of traditional TFM regularization schemes and the lack of posterior uncertainty quantification, we developed TFM-UQ – a hierarchical Bayesian framework with Markov Chain Monte Carlo (MCMC) sampling and non-informative hyperprior distributions for the hyperparameters. TFM-UQ accounts for the measured PIV uncertainty when enforcing prior constraints, such as the smoothness of the traction stress field, thereby adjusting the level of regularization to the spatial distribution of the measurement errors. We showed that this approach locally balances overfitting and smoothing, and can outperform traditional regularization methods, especially in experiments where the quality of the fluorescent bead images or the measured displacements exhibit appreciable gradients.

Uncertainty propagation in TFM-UQ considers the amplification of high-frequency noise in the substrate displacements and the effect of the regularizing prior(s) defined to suppress that noise. As a consequence of the interplay between these factors, the bounds of $\sigma_t$ are determined by the PIV uncertainty and the strength of the regularizer respectively. This interdependence reflects the fact that regions with higher measurement uncertainty are regularized more heavily, so the prior information of the regularizer overrides the PIV contribution in those regions. The analysis also showed that the posterior traction precision is higher for softer gels in the long wavelength (pure shear) limit. The hierarchical formulation effectively applies a Student-t marginal prior that has longer tails than a Gaussian distribution, however the shrinkage in parameter complexity enters only as an average ($||Lt||_2$) suppressing sharp traction values. Application of TFM-UQ to highly resolved displacement data such as DNA fluorocubes [83] could benefit from such adaptive and non-Gaussian prior distribution.

Future studies should address the interactions between heteroskedastic measurements and Bayesian priors in the context of TFM and the effect of substrate material parameters on the uncertainty propagation. In particular, the possibility of introducing non-global smoothness priors (locally adaptive shrinkage or heteroskedastic priors) deserves further investigation. While TFM-UQ improves the method by modeling the image-based deformation (i.e., PIV-UQ) uncertainty as an additive error, the experimental parameters that enter the transformation $M$ (such as Young's modulus of the gel that could exhibit local stiffening) are not explicitly modeled. Instead, they are lumped into $\varepsilon_\beta$ as a global error. The possibility of extending non-zero covariance structure to model error from empirical data [84] or incorporating multiplicative noise for variables affecting $M$ warrant future investigation.

## 5. Conclusion

This manuscript presents a traction force microscopy technique that can estimate the measurement uncertainty associated with image quality and critical analysis parameters such as regularization of the recovered traction stresses. A noteworthy contribution of this work is the investigation of error bounds for the magnitude and direction of the traction stress vector measured at each point in space and time of an experiment. This feature of uncertainty quantification facilitates the standardization of TFM protocols, distinguishing biological heterogeneity from TFM measurement confidence, automating quality assessment, applying spatially adaptive smoothing according to local data quality. Moreover it provides a framework to quantify the effects of parameters such as regularization, PIV window size, and image processing on the precision of TFM. We anticipate these new capabilities will pave the way for more robust, high-throughput applications of TFM that can bypass user intervention for every image.

TFM-UQ can run on a personal computer in a few minutes depending on the resolution. The requirement of repeated PIV runs and solution of elastostatic equations for bootstrap and MCMC respectively comes at a higher cost compared to a traditional TFM implementation. For images of uniformly high quality, traditional TFM can produce similar traction maps with lower computational demand, albeit without knowledge of the confidence levels. If runtime is not a limiting factor, TFM-UQ is recommended for added insights. The computational time is expected to reach near real-time with additional investigations of variational methods or approximate inference methods and improvement in computational power of consumer-grade hardware such as GPUs.

## Supporting information

**S1 Text.** Supplementary information.
(PDF)

## Acknowledgments

We thank Dr. Alejandro Gonzalo and Dr. Susanne Rafelski for providing critical feedback on the manuscript, and Dr. Kelly Stevens for providing C3H/10T1/2 fibroblast cell line.

## Author contributions

**Conceptualization:** Adithan Kandasamy, Juan Carlos del Alamo.

**Formal analysis:** Adithan Kandasamy.

**Funding acquisition:** Mark Mercola, Juan Carlos del Alamo.

**Investigation:** Adithan Kandasamy, Yi-Ting Yeh.

**Project administration:** Yi-Ting Yeh, Mark Mercola, Juan Carlos del Alamo.

**Software:** Adithan Kandasamy, Ricardo Serrano.

**Supervision:** Mark Mercola, Juan Carlos del Alamo.

**Writing – original draft:** Adithan Kandasamy, Yi-Ting Yeh, Ricardo Serrano, Mark Mercola, Juan Carlos del Alamo.

**Writing – review & editing:** Adithan Kandasamy, Yi-Ting Yeh, Ricardo Serrano, Mark Mercola, Juan Carlos del Alamo.

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
