## [Decision Letter · Decision Letter 0]

2 Sep 2024

Dear Dr. del Alamo,

Thank you very much for submitting your manuscript "Uncertainty-Aware Traction Force Microscopy" for consideration at PLOS Computational Biology.

As with all papers reviewed by the journal, your manuscript was reviewed by members of the editorial board and by several independent reviewers. In light of the reviews (below this email), we would like to invite the resubmission of a significantly-revised version that takes into account the reviewers' comments.

We cannot make any decision about publication until we have seen the revised manuscript and your response to the reviewers' comments. Your revised manuscript is also likely to be sent to reviewers for further evaluation.

Sincerely,

Daniel A Beard

Section Editor

PLOS Computational Biology

Daniel Beard

Section Editor

PLOS Computational Biology

Reviewer's Responses to Questions

**Comments to the Authors:**

Reviewer #1: This is a method development manuscript on the topic of traction force microscopy (TFM), which is a method to quantify traction forces produced by cells by measuring displacements in the substrate beneath the cells. There are multiple technical challenges in TFM, with variable image quality and sensitivity to noise being major challenges. The main contribution of this manuscript is that it describes methods to handle those common challenges. The approach used here is to quantify uncertainty in the measured substrate displacements, which in turn is used to determine the optimal regularization/smoothing parameters used in the calculation of traction forces. In this approach, the regularization parameters can vary over space. Questions about the manuscript are listed below, approximately in order of importance.

1. The first, and most important, comment is about how the authors envision this method being used. The introduction (line 70 of page 4) suggests that a reason that TFM is not commonly used is that it is technically challenging: “perpetuating the view that TFM is technically involved and challenging to interpret, hindering its more widespread adoption.” The implication here is that TFM is not widely used because the technical details are complex. The present manuscript presents a method for TFM that is substantially more complex than the commonly used methods. It would seem that this added complexity would further hinder adoption of TFM, unless the authors are proposing that TFM calculations become a black box, with no need for the user to understand the process. For the method described in this manuscript, the black box approach would be highly problematic, because the regularization parameters can vary over space due to changes in image quality, which in turn affects the traction forces output by the computations. If a user were unaware that the regularization varies in space depending on image quality, they could mistakenly infer changes in the traction data as being the result of biological changes rather than variation in the image quality. For this reason, a black box approach is not possible, and the users will require some understanding of the details of how the method is implemented. Thus, the notion that this method could widen the user base for TFM seems to be incorrect. With this in mind, the authors should consider who is the ideal user for this method – what expertise is needed and what type of experiment would benefit from this method? And what type of experiment could be successfully done with simpler implementations of TFM? These details should be explained in the introduction and discussion/conclusions.

2. Figure 7 shows a representative result of the method wherein the image has lower quality on the right side, compared to the left, which creates a greater uncertainty on the right side compared to left. As a result, the adaptive TFM approach described here uses a higher value of the regularization parameter on the right side compared to left. Does this result in a lower magnitude of recovered traction on the right side of the image compared to left? If yes, it would see like this causes substantial challenges in interpreting the data – here a change in tractions can result from either a change in force produced by the cell or a change in regularization parameter. In other implementations of TFM, the regularization parameter is kept constant. In those other implementations, an experienced user might be able to see anomalies in the recovered tractions due to gradients in image quality and exclude that image from the data set. Here, the variable regularization seems to obscure effects of variable image quality, which could lead the user to conclude, incorrectly, that the magnitude of cell traction varies over space. How can the user of this method resolve this challenge that in this method, the tractions output are a combination of the forces produced by the cells and the spatial variability of the regularization parameter?

3. To quantify uncertainty in the displacement field recovered by particle image velocimetry (PIV), the authors propose a bootstrapping approach, wherein pixels in a PIV window are sampled randomly with replacement, which creates a bootstrapped window that is used in the PIV code for estimating the displacement of that window. This approach seems to be problematic for two reasons. First, the spatial intensity of the pixels in the window are correlated in space, because a fluorescent particle imaged by the microscopy is typically larger than a single pixel. Doesn’t the bootstrap approach assume that the samples are independent? Second, this bootstrapping approach does not seem to be a physically meaningful means of representing noise that can occur during imaging. The boostrapping leaves some pixels blank (with values of 0), which does not occur when imaging, unless there’s a dead pixel in the camera. It seems that the bootstrapping approach is not a very good choice for estimating uncertainty in the PIV data. Would it be better to take a different approach? One idea is repeating the analysis after adding noise to the PIV window, though maybe the authors can think of a better method. Alternatively, if the bootstrapping is the best approach, the authors must describe in the manuscript why the two concerns above are not relevant.

4. The manuscript shows some comparisons to Tikhonov regularization, which is one of the common approaches to handle noise. Another common approach is to use a constraint rather than regularization. A simple and physically meaningful method for constraint is to use the displacement field as a boundary condition in a finite element model for the substrate – this constraint enforces both conservation of momentum and geometric compatibility. An advantage of this constraint-based approach is that there is no need to choose a regularization parameter. How does the author’s method compare to a constrained case such as this one?

5. Figures 5 and 6 show synthetic images of fluorescent particles in a substrate that are used for testing of the method. From the images, it appears that the densities of fluorescent particles are rather sparse. It is well known in image correlation methods like PIV that a denser distribution of speckles will create better measurements with higher spatial resolution and/or lower noise. Can the authors repeat the procedure using images that have a denser distribution of particles and compare the results using both their proposed method and other common methods for TFM?

6. The first paragraph of p. 27 describes how the fluid mechanics community has developed the method referred to as PIV. The authors should mention the existence of other communities – the solid mechanics community uses the term “digital image correlation” and the computer vision community uses the term “optical flow.” Details of the implementation of PIV, DIC, and optical flow can vary, but these different communities all have the same goal of obtaining kinematic data from images, so it is important to acknowledge that all communities have contributed.

Reviewer #2: The authors present the first traction force microscopy (TFM) study that explicitly addresses the local uncertainty in the traction stress reconstruction. In line with similar efforts in the data sciences, they call their method “uncertainty aware TFM” and use the abbreviation “TFM-UQ” (for uncertainty quantification). In principle earlier TFM-methods based on Bayesian statistics (e.g. Refs. 9 or 21) already contained such a possibility, but it was never investigated before in the same depth as done here. Thus, this work is truly novel and fills an important gap in the TFM-field. The manuscript is well written, albeit very technical. The motivation for this work is nicely laid out and then the new methods are described, first the uncertainty treatment for PIV (using a bootstrap method) and then for TFM itself (using a hierarchical Bayesian scheme). Both methods are validated first with simulated data and then with experimental cell data. Here micropatterning and the image correction of the Zeiss microscope are being used to check the disentangle the effect of imaging and TFM. It is shown that poor PIV results for the raw image, and that poor TFM-reconstruction results at low traction. As far as I can see, all these results are consistent and convincing.

Overall, this work is a nice advance in the TFM-field and I recommend to proceed towards publication. However, there are a few issues one should address before making a final recommendation.

Major comments

This manuscript is rather technical in nature and would also fit to a journal on engineering, applied math or data sciences, at least in regard to the methods part. To improve the fit to PLOS CB, I suggest two measures. First, I note that on page 13, the computing time is recorded, but the authors should also explain in which language their code is written (Matlab, Cpp, Python ?) and which libraries are being used. Also, the code is not provided yet. This makes it difficult to assess how well this code can be used by the general reader. I suggest to include more comments on the code and how to use it, to make it a useful resource for readers who want to apply these advances.

Second, I think that the life science community might want to learn more about the sources of uncertainty from the biological side. I encourage the authors to consider to include some experiments on different cell lines or inhibition experiments, and to show under which biological conditions uncertainty arises. Because the authors focus on the technical aspects, the biological aspects could be strengthened.

It is a well-known riddle in the field that DNA-sensor based measurements of traction forces give much more heterogeneous results, with some bonds experiencing exceedingly large forces (like 50 pN versus 5 pN baseline). However, it is not clear if these are artifacts of the method or real results. Does TFM-UQ give any hints to such large local forces? I suspect that the Gaussian assumption excludes statements on extreme values. Please explain.

The authors assume “displacement noise”, compare Eq. 2. They differ between two sources for noise, the uncertainty of the image processing which they can infer from the PIV and which might result from e.g. aggregates of fiducial beads; and other sources, like the elastostatic model. Is it clear that imaging really leads to additive noise in the displacement, for example if one could simulate the complete process of image generation? Similar for the mechanical noise: if the material law was not correct or if there were local heterogeneities in the polymer gel, would that also lead to an additive noise term in the displacement? In addition, I wonder if the two noises might be connected, like material heterogeneities leading to both imaging and mechanical noise? It would be interesting to learn if the authors have some thoughts on this set of questions.

Minor comments

Fig. 5: it is not clear to me how large the ensembles are and if the TFM-UQ also averages over the same ensembles? For a TFM-experiment, one has only one shot and it was my understanding that TFM-UQ uses only one image pair for bootstrapping. Please clarify.

Fig. 6C: coloring and legend are not clear, should this not be a horizontal gradient?

**Have the authors made all data and (if applicable) computational code underlying the findings in their manuscript fully available?**

Reviewer #1: Yes

Reviewer #2: **No: **The Github link does not work yet and the code is not described, see my report.

PLOS authors have the option to publish the peer review history of their article (what does this mean?). If published, this will include your full peer review and any attached files.

Reviewer #1: No

Reviewer #2: No
---

## [Decision Letter · Decision Letter 1]

23 Apr 2025

Dear Dr. del Alamo,

We are pleased to inform you that your manuscript 'Uncertainty-Aware Traction Force Microscopy' has been provisionally accepted for publication in PLOS Computational Biology.

Best regards,

Daniel A Beard

Section Editor

PLOS Computational Biology

Daniel Beard

Section Editor

PLOS Computational Biology

Reviewer's Responses to Questions

**Comments to the Authors:**

Reviewer #1: It is clear that the authors carefully considered my comments. Some changes/additions to the manuscript have been made. When the authors didn't agree with changes I suggested, they gave a reasonable explanation for their thinking. I have no further questions or comments.

Reviewer #2: The authors have responded very well to my comments. In particular, they have followed my two main suggestions by adding new results. (1) They now provide their computer code as Matlab files accessible on GitHub. I have checked the code, it is transparent and runs ad hoc with one example given. The drawback of using Matlab is of course that users need a Matlab version. But given that, any user can now run the code and get results by reading in own image files. (2) To address the question of biological variability, the authors now supply Fig SI.6 for cells on micropatterns of different sizes and essentially demonstrating that larger forces on large islands come with larger uncertainties. In the future, this could be used to compare different biological situations of interest. My other comments have also been addressed very well by making corresponding changes to the text.

In my first report, I recommended to proceed towards publication, and with the revised version, acceptance can now be recommended. For the final version, in new Fig SI.6, uM should be replaced by um (twice). I am also a bit puzzled because the uncertainty cones shown in the cartoon of Fig 1 are not given in the results, only the scalar uncertainty; that is completely fine with me and I assume that the authors could plot the cones if needed for a specific context. Finally, the gradient in Fig 6C seems to be biphasic in my PDF, which does not fit to Fig 6D, where noise is monotonously increasing to the right; maybe a final version could make this more compatible.

**Have the authors made all data and (if applicable) computational code underlying the findings in their manuscript fully available?**

Reviewer #1: None

Reviewer #2: Yes

PLOS authors have the option to publish the peer review history of their article (what does this mean?). If published, this will include your full peer review and any attached files.

Reviewer #1: No

Reviewer #2: **Yes: **Ulrich Schwarz

---

## [Editor Report · Acceptance letter]

PCOMPBIOL-D-24-01236R1

Uncertainty-Aware Traction Force Microscopy

Dear Dr del Alamo,

I am pleased to inform you that your manuscript has been formally accepted for publication in PLOS Computational Biology. Your manuscript is now with our production department and you will be notified of the publication date in due course.

With kind regards,

Anita Estes
